# VAE with Hyperspherical Coordinates: Improving Anomaly Detection from Hypervolume-Compressed Latent Space

## Abstract

Variational autoencoders (VAE) encode data into lower-dimensional latent vectors before decoding those vectors back to data. Once trained, one can hope to detect out-of-distribution (abnormal) latent vectors, but several issues arise when the latent space is high dimensional. This includes an exponential growth of the hypervolume with the dimension, which severely affects the generative capacity of the VAE. In this paper, we draw insights from high dimensional statistics: in these regimes, the latent vectors of a standard VAE are distributed on the 'equators' of a hypersphere, challenging the detection of anomalies. We propose to formulate the latent variables of a VAE using hyperspherical coordinates, which allows compressing the latent vectors towards a given direction on the hypersphere, thereby allowing for a more expressive approximate posterior. We show that this improves both the fully unsupervised and semi-supervised anomaly detection ability of the VAE, achieving the best performance on the datasets we considered, outperforming existing methods. For the unsupervised and semi-supervised modalities, respectively, these are: i) detecting unusual landscape from the Mars Rover camera and unusual Galaxies from ground based imagery (complex, real world datasets); ii) standard benchmarks like Cifar10 and subsets of ImageNet as the in-distribution (ID) class.

## 1 Introduction

Anomaly detection (AD) can be done in a fully unsupervised or semi-supervised manner. Fully unsupervised anomaly detection, where one *does not have access to sub-class information/labels in the normal/ID data*, is a challenging task, and two main approaches have shown promise. They both rely on an autoencoder (AE), which encodes the data into a lower dimension latent space, before decoding the latent vectors back to data. The main assumption is that the AE, having learned to encode/decode the training dataset, would do poorly in processing a sample outside that distribution.

The detection can either be done by comparing the reconstructed and the original data Dietterich (2021); Kerner et al. (2020), or by detecting whether a latent vectors lie outside the "normal" *latent* distribution. A variation of the former uses a diffusion-based generative model to reconstruct data, but the detection is also between the generated and original data Liu et al. (2025).

Many recent works have focused on the semi-supervised AD case, where one uses sub-class information from the normal training set to help the task, for example by disentangling the normal subclasses with a classifier, e.g., a ResNet Wang et al. (2023); Li et al. (2025); the 'good' embeddings/features from the penultimate layer can then be used for AD via some standard method outlier detection techniques, e.g., k-NN Sun et al. (2022). This greatly simplifies the problem.

In this work, we explored the performance of our method on both of these two types of AD scenarios.

Our main contribution is a novel way to detect anomalies in latent space. AD is challenging due to the high dimensionality of practical datasets such as the images used in this paper: because of concentration of measure effects, when encoding samples uniformly to a latent hypersphere, they tend to *only* populate its '*equators*'. We proposed to convert the latent vectors from Cartesian coordinates to hyperspherical coordinates, de facto disentangling the dimensions: a point on a hypersphere can

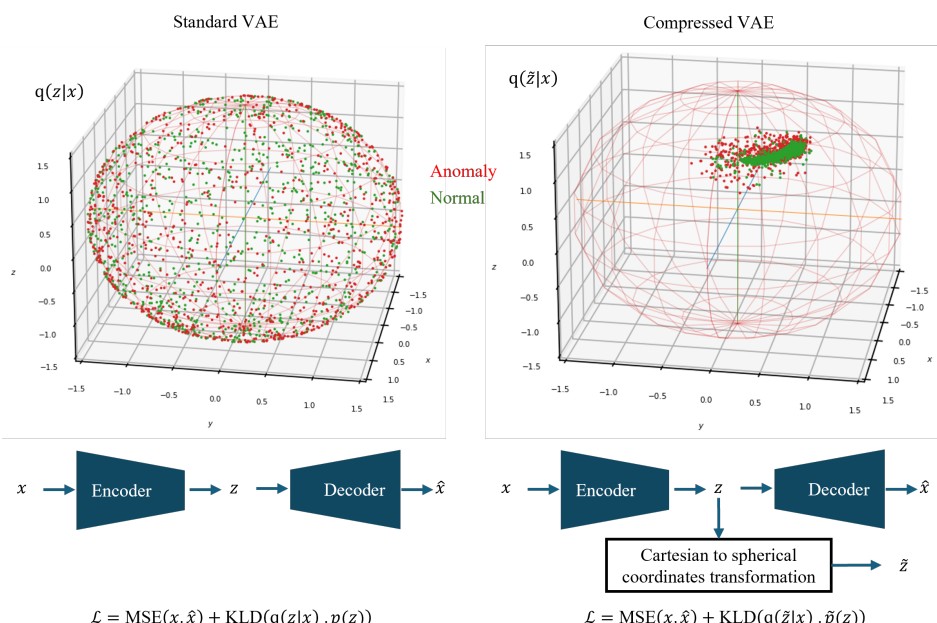

$$\mathcal{L} = \text{MSE}(x, \hat{x}) + \text{KLD}(\text{q}(z|x), p(z)) \qquad \mathcal{L} = \text{MSE}(x, \hat{x}) + \text{KLD}(\text{q}(\tilde{z}|x), \tilde{p}(z))$$

Figure 1: Proposed method for the **fully unsupervised case**. The standard VAE (left) is modified (right) by converting the latent vectors into hyperspherical coordinates. In our new formulation, the latent vectors from the normal class in green can be moved during training towards a given direction on the hypersphere, forming a dense and compact "island", illustrated here by projecting the latent distributions on a 2D sphere (see subsection Results 4.3 for more details about how this is done). Anomalies in red are detected by measuring their distance to the island. The figure corresponds to results from the experiment on the Galaxy Zoo dataset (cf. Table 1, third column).

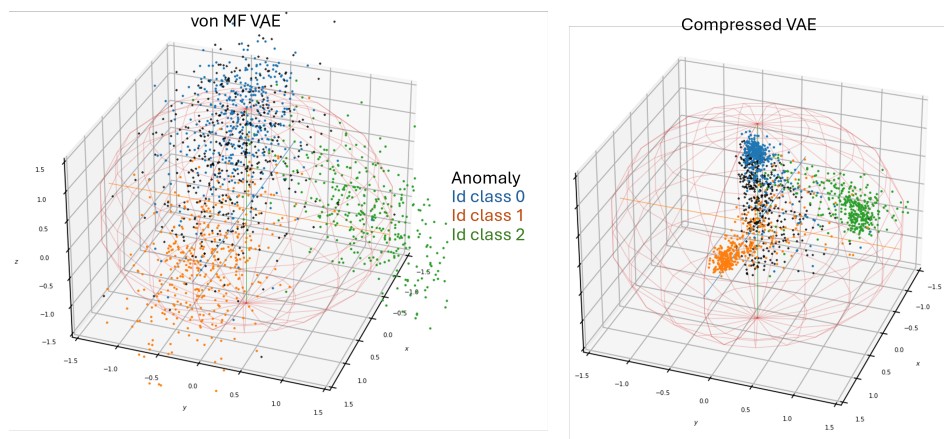

Figure 2: Proposed method for the **semi-supervised case**. In this case, a compression of the same type as in the previous figure is done on each of the ID class clusters, simply by re-orienting the full hyperspherical coordinate system such that the first angular coordinate is the angle w.r.t. the Cartesian orthogonal axis whose number is equal to a corresponding ID class label. The von Mises-Fisher-based method (von MF) shows more noisy and dispersed samples because having only one single parameter (the first hyperspherical angle) to compress the volume and thus reduce the sparsity of the HD space is not enough, as we show in the Supplementary materials (Supp.), where we also see that t-SNE can be misleading for assessing compression. In contrast, our method compresses all of the hyperspherical angles. The figure corresponds to results from the experiment on Imagenette vs close Imagenet (cf. Table 3).

be moved on the surface by changing only one hyperspherical angle. This is impossible to do using standard Cartesian coordinates: moving a point on the hypersphere surface involves modifying every Cartesian latent dimension. Because of that disentanglement, we could reformulate the VAE cost function to move all the latent samples towards a given direction on the hypersphere, creating a very dense island, away from the high hypervolume equators. Measuring how far a sample is from that island of normal data becomes easier than in the case of a uniform distribution on the hypersphere with its vast equators, and a simple $k-$nearest neighbors achieves better results for AD than existing methods. The creation of a dense island by this particular method has already been shown to be useful when using the VAE for the generative task Ascárate et al. (2025), and here we show its value for AD.

Reviews on the standard VAE can be found in Kingma & Welling (2014; 2019). The latent space tends towards a high dimensional independent multivariate Gaussian, which has properties that we briefly review next.

## 1.1 HIGH DIMENSIONAL SPACES

Sampling from a multivariate Gaussian in a high-dimensional (HD) Euclidean space of dimension $n$ exhibits several counterintuitive properties. Although the origin has the highest probability density, the probability of drawing samples near it is nearly zero. Instead, most samples concentrate near the $(n-1)$-dimensional hypersphere $\mathbb{S}^{n-1}_{\sqrt{n}}$ of radius $\sqrt{n}$. The norm of the samples follows a $\chi(n)$ distribution, which implies that samples lie within a thin shell around the hypersphere. The thickness of this shell relative to its radius $\sqrt{n}$ shrinks as $n$ increases.

As $n$ grows large, the distribution of $\mathcal{N}(0, I_n)$ approaches the uniform distribution on the hypersphere. Furthermore, any two independent samples from $\mathcal{N}(0, I_n)$ are always nearly orthogonal to one another, a property called *almost-orthogonality* (see Vershynin (2018) for a formal treatment).

These behaviors are deeply tied to how (hyper-)volume behaves in HD spaces. Under the uniform measure on a hypersphere, most of its exponentially growing volume in $n$ is concentrated in extremely thin **equatorial** bands relative to **any randomly chosen "north pole."** This is a truly remarkable and deep fact. It is formalized in Wainwright (2019), and is part of the broader notion known in mathematics as *concentration of measure*. Standard low-dimensional (2D or 3D) intuitions about spheres break down in HD spaces, and such properties significantly affect anomaly detection in models like VAEs, as we discuss next. We give some simple yet insightful examples in Supp., as well as some remarks on its connection with the volume, which will be key in our paper.

## 1.2 ANOMALY DETECTION IN HIGH DIMENSION

High resolution/complex images need many latent dimensions to capture all the information they convey. But, as mentioned before, HD spaces often display properties that go against the intuition gained from their low dimensional counterparts, where many of the original methods for AD were developed.

For example, a common assumption is that anomalies will be located in the tail of the normal data distribution. Then, for an AD method to have good performance, one would need this tail to allow for some concentration of samples, that is, a heavy tail. In HD spaces, the tails of a wide class of functions of distributions, like the norm of a sample from a standard Gaussian, tend to be very short. This is the classic concentration of measure phenomenon. More formally:

**Proposition** (Measure Concentration Vershynin (2018); Wainwright (2019); Akers et al. (2024)): Let $z$ be a Gaussian random vector and $f : \mathbb{R}^n \longrightarrow \mathbb{R}$ a Lipschitz function with Lipschitz constant $K$. Then,

$$\Pr\left(\mid f(z) - \mathbb{E}f(z) \mid \geq t\right) \leq 2\exp\left(-\frac{t^2}{4K}\right). \square$$

Note that the previous statement *does not* depend on the dimension $n$ (see also Akers et al. (2024), Appendix B, for a compact introduction to the general Riemannian case). The effects of HD can be seen when selecting a particular function: e.g., for the previously alluded concentration of the norm of the Gaussian around $\sqrt{n}$, this follows from the general result applied to $f(z) = \parallel z \parallel$, since $\mathbb{E} \parallel z \parallel \sim \sqrt{n}$ and $K = 1$.

This can affect the standard VAE both as a generative model Ascárate et al. (2025) as well as a tool for AD Tam & Dunson (2025), since it assumes a standard Gaussian distribution as prior. *Crucially, the concentration effects will affect the anomaly score itself*, since it is a function from the Gaussian-like latent encodings to the reals, and thus produce short tails in its distribution, which makes the disentangling between the normal and abnormal classes more difficult (see Supp. for examples from our experiments).

Our main hypothesis is that a model with a HD latent distribution/representation resembling a HD Gaussian will be very negatively affected by the concentration of measure phenomena for tasks like AD.

## 2 METHOD

### 2.1 VAE WITH HYPERSPHERICAL COORDINATES

Our approach is based on formulating the initial KL divergence term with a prior from the original VAE, which is in Cartesian coordinates, to one in hyperspherical coordinates. See Supp. for the standard conversion formulas between Cartesian and hyperspherical in high dimension.

In Cartesian coordinates, the KL divergence between the estimated posterior defined by $\mu_k$ and $\sigma_k$, and the prior defined by $\mu_k^p$ and $\sigma_k^p$ has been well documented Kingma & Welling (2014). It can be written as,

$$\text{KLD}_{\text{CartCoords}}^{w/Prior} \simeq \sum_{k=1}^{n} \left( (\mathbb{E}_b[\sigma_k] - \sigma_k^p)^2 + \sigma_b[\sigma_k]^2 + (\mathbb{E}_b[\mu_k] - \mu_k^p)^2 + \sigma_b[\mu_k]^2 \right), \quad (1)$$

where $\mathbb{E}_b$ and $\sigma_b$ denote the mini batch statistics of size $N_b$.

This formulation using Cartesian coordinates includes batch statistics and was partly inspired by the construction in Bardes et al. (2022). It will be useful for our next step.

We now introduce hyperspherical coordinates in the KLD formulation. We start with the Cartesian coordinates $(\mu_i, \sigma_i)$, given by the encoder, and transform these to their hyperspherical counterparts $(\overset{\mu}{r}, \overset{\mu}{\varphi}_k; \overset{\sigma}{r}, \overset{\sigma}{\varphi}_k)$ with $r$ a scalar and $k$ the index of the $n-1$ spherical angles.

The KLD-like objective becomes for the angles $\varphi_k$,

$$\text{KLD}_{\text{HSphCoords}}^{w/Prior}(\varphi_k) = \sum_{k=1}^{n-1} \left( \alpha_{\sigma,k} \left( \mathbb{E}_b[\cos \overset{\sigma}{\varphi}_k] - a_{\sigma,k} \right)^2 + \beta_{\sigma,k} \left( \sigma_b[\cos \overset{\sigma}{\varphi}_k] - b_{\sigma,k} \right)^2 \right.$$
$$\left. + \alpha_{\mu,k} \left( \mathbb{E}_b[\cos \overset{\mu}{\varphi}_k] - a_{\mu,k} \right)^2 + \beta_{\mu,k} \left( \sigma_b[\cos \overset{\mu}{\varphi}_k] - b_{\mu,k} \right)^2 \right), \quad (2)$$

and for the norm $r$,

$$\text{KLD}_{\text{HSphCoords}}^{w/Prior}(r) = \alpha_{\sigma,r} \left( \mathbb{E}_b[\overset{\sigma}{r}] - a_{\sigma,r} \right)^2 + \beta_{\sigma,r} \left( \sigma_b[\overset{\sigma}{r}] - b_{\sigma,r} \right)^2$$
$$+ \alpha_{\mu,r} \left( \mathbb{E}_b[\overset{\mu}{r}] - a_{\mu,r} \right)^2 + \beta_{\mu,r} \left( \sigma_b[\overset{\mu}{r}] - b_{\mu,r} \right)^2, \quad (3)$$

with the priors for the mean over the batch $a_{i,j}$, the standard deviation over the batch $b_{i,j}$, and the gains for each term $\alpha_{i,j}$, $\beta_{i,j}$, for $i \in \{\sigma, \mu\}$ and $j \in \{1, ..., n-1, r\}$.

We use the cosines rather than the angles to avoid costly extra computations of the corresponding arccosines. The coordinate transformation is done using a vectorized implementation (code provided in Supp.). The reparameterization trick is still done in the Cartesian coordinates representation. The total loss is,

$$\mathcal{L} = \text{MSE}(x, \hat{x}_z) + \beta \left( \text{KLD}_{\text{HSphCoords}}^{w/Prior}(\varphi_k) + \text{KLD}_{\text{HSphCoords}}^{w/Prior}(r) \right) \quad (4)$$

## 2.2 VOLUME COMPRESSION OF THE LATENT MANIFOLD

We discussed previously that the standard VAE forces the latent samples to be uniformly distributed on the hypersphere, which in high dimensions results in data located within equators of the hypersphere where the volume is the greatest. A benefit of using hyperspherical coordinates is the ability to set a prior for the $\varphi_k$ that forces the latent samples away from the equators of *each* and *all* $(n-k)$-hyperspheres, $\forall k$, contained (or equal, if $k = 1$) in the initial one (since they are HD too), thereby escaping these regions. This can be done for each angular coordinate, as all are uncorrelated with each other, by simply setting (north pole),

$$\boxed{a_{\mu,k} = 1, \ \forall k}. \tag{5}$$

By doing so, the samples can be moved to a zone with much reduced volume. We speculate that this allows for a more expressive approximate posterior, better suited for the AD task than the one obtained from a standard Gaussian as prior. See Supp. for a more detailed analysis of the behavior of the hypervolume in this situation. Finally, by setting,

$$a_{\mu,r} = \sqrt{n}, \tag{6}$$

(and normalizing $z$, after sampling via the reparameterization trick, to the same radius $\sqrt{n}$) we can force the latent samples to be on the hyperspherical surface of that radius.

For the fully unsupervised case we compress all of the normal samples into a same cluster. For the semi-supervised case, we re-orient the full hyperspherical coordinate system such that the first angular coordinate is the angle w.r.t. the Cartesian orthogonal axis whose number is equal to a corresponding ID class label; thus, we get a compact cluster for each normal class close to the intersection between the hypersphere and the corresponding labeled axis. This re-orientation is done by applying the roll operation from PyTorch on each latent representation, where the amount of shifting is equal to the class label of the sample (in our implementation we spaced each of the labeled dimensions by 10 unlabeled ones for having a more convenient way to see the stacked histograms in Supp.).

## 2.3 VON MISES-FISHER-BASED METHODS ARE A SUBSET OF OUR GENERAL APPROACH

The case when one reduces *only* a single angular coordinate, e.g., $\varphi_1$, corresponds exactly to varying the *single scalar variance parameter* on a von Mises-Fisher distribution on the hypersphere, which is defined as an isotropic (hence the single free parameter rather than a vector or matrix for this variance) Gaussian whose domain is restricted to the hypersphere. This approach has been used recently in the literature when dealing with encodings into the hypersphere for the semi-supervised modality Ming et al. (2023); Ghosal et al. (2024), but as we discuss in more detail in Supp., it cannot reduce the hypervolume as fast as our method in case one wishes to do so.; hence, it will always have more internal dispersion compared to our approach.

## 2.4 ANOMALY DETECTION

AD methods can use a distance function $d : Z \times Z \longrightarrow \mathbb{R}^+$ on the space of data points $z \in Z$ as the primary tool to define and compute the anomaly score. This is usually a high-dimensional Euclidean space or a hyper-surface; the distance function can be either the Euclidean distance or the corresponding induced distance, respectively.

The simplest of such methods is the $k-$NN score, where the anomaly value of a query data point $z$ is defined as the mean distance to its $k-$nearest neighbors, that is,

$$A(z) = \frac{1}{k} \sum_{i=1}^{k} d(z, z_i),$$

where the index $i$ refers to an ordering of the points in $Z$ such that $d(z, z_1) \leq ... \leq d(z, z_n)$.

In our case, after training the VAE with only the nominal data $X$, we encode this data into the latent space $Z$ and consider the corresponding means $\mu_x$, $x \in X$, given by the encoder. During test time,

given a query point $x_{test}$, we encode it to obtain the mean $\mu_{x_{test}}$ and compute its anomaly score $A(\mu_{x_{test}})$ via the previous $k-$NN score w.r.t. all the mentioned means $\mu_x$ of the training set.

We use the standard Euclidean distance and set $k = 3$.

# 3 RELATED WORK

The Isolation Forest Liu et al. (2008); Emmott et al. (2013) (iForest or IF) method creates a forest of random axis-parallel projection trees. It derives a score based on the observation that points becoming isolated and closer to the root of a tree are easier to separate from the rest of the data and therefore, are more likely to be anomalous.

When working with VAEs, there is a natural method that suggests itself for AD. This relies on the hypothesis that the reconstruction, by the trained network, of data close to the training set should be reasonably good, since it was optimized for that task, while the reconstruction for anomalies should be worse, since these data points are intrinsically different than those in the training set, the network should have more problems reconstructing them Pang et al. (2022); Dietterich (2021). A common anomaly score is thus the reconstruction error (MSE): $A(x) =\parallel x - \hat{x} \parallel$.

It is hard to prevent a plain autoencoder from learning to be a general image compression algorithm, though. When this occurs, it is not helpful for anomaly detection via reconstruction error, because it does not fail on new images Bouman & Heskes (2025); Gong et al. (2019).

Regarding the novel method for AD that we present in the next section, a recent work Fu et al. (2024) shares some similarities. The main differences are: i) volume compression is done radially between all points, which, from our view, misses the key point of the peculiar angular-like, equatorial distribution of volume in the HD regime; ii) the method is specifically designed for AD only, with only an encoder network for feature extraction, while our method uses a VAE, a generative model, which then can be used for other tasks, e.g., for improving generation, as in Ascárate et al. (2025); iii) experiments used only simulated anomalies, done with classification datasets like CIFAR10 in which a class is the anomaly to the other nine. We found this latter aspect problematic, since these fictional AD scenarios do not seem adequate for evaluation of fully unsupervised AD methods, as we will argue in more detail in Supp. The method in this reference, like ours, uses a $k-$NN approach Dietterich (2021) for performing AD. Unfortunately, the code is not provided.

In the semi-supervised realm, there are a series of recent methods Sun et al. (2022); Ming et al. (2023); Ghosal et al. (2024) based on performing AD on the feature space of a deep classifier, e.g., a ResNet, via the $k-$NN approach. The work Ming et al. (2023), in particular, uses hyperspherical embeddings via clusters modeled by von Mises-Fisher (vMF) distributions and $k-$NN on that set-up for AD; thus, it is, conceptually, the closest to our approach. We did not found references using the vMF approach for the fully unsupervised case.

For the case of our ImageNet-based set-up (Imagenette vs. close Imagenet), we implemented the idea in Sun et al. (2022) from scratch, with the same ResNet as our encoder in the AE and VAE, and is reported as KNN* in Table 233 (as emphasized in those references, we normalize the test set before performing the KNN). Furthermore, our Comp.VAE method, when restricted to compress only the first hyperspherical angular coordinate (see Method section), is identical to a von Mises-Fisher method. Thus, this provides a straightforward comparison with that idea, since everything else remains identical in the setup w.r.t. the full Comp.VAE.

# 4 EXPERIMENTAL RESULTS

## 4.1 MODEL AND IMPLEMENTATION

For all our experiments, we use a small, and customized for serving in a VAE, standard ResNet-18-like architecture He et al. (2016) for both encoder and decoder, for a total of around 0.1 to 1 million parameters. When using the loss in hyperspherical coordinates (4), we use an annealing-like schedule Fu et al. (2019) for the gain $\beta$ of the KLD-like loss, which simply increases proportionally with $\sqrt{\text{epoch}}$ for a total of 100 epochs. See Supp. for more details.

## 4.2 DATASETS

**Fully unsupervised case**    The Mars Rover Mastcam dataset Kerner et al. (2020) for AD comprises multispectral images from the rover-based planetary exploration missions on the planet Mars. The training, all normal, consists in 9124 images of size $64 \times 64$, and 6 channels (i.e., multispectral imaging). See Supp. for more details.

The Galaxy Zoo dataset Lintott et al. (2008; 2011) for AD covers 61578 galaxies, each represented by a $400 \times 400$ sized image with 3 channels. The galaxies were classified by volunteers using a series of questions. One of the questions (corresponding to Class 6.1) "Is there anything "odd" about the galaxy?", can be used as a ground truth for anomalies. Following Lochner & Bassett (2021), we extracted all objects with a Class 6.1 score greater than 0.9, which means at least 90 per cent of the volunteers labeled the galaxy as odd. This results in 924 anomalies. Then we randomly selected 924 images from the remaining ones to build the normal part of the test set. Thus, we get a training set of 59730 normal images and a test set of 1848, half of them normal and the other half abnormal. We resized the dataset to $64 \times 64$ images in order to make the training less expensive to run. See Supp. for more details.

The popular MVTec Bergmann et al. (2021) dataset for AD is unfortunately too small in our view (less than 5000 training samples), and therefore we did not consider it in our study.

**Semi-supervised case**    We follow common practice and use **CIFAR-10** (10 classes) as in-distribution (ID) dataset (Krizhevsky & Hinton, 2009). For far out-of-distribution (far-OOD) evaluation we use six widely adopted datasets, all resized to $32 \times 32$: **Textures** (Cimpoi et al., 2014), **SVHN** (Netzer et al., 2011), **LSUN-Crop** and **LSUN-Resize** (Yu et al., 2015), **iSUN** (Xu et al., 2015), and **Places365** (Zhou et al., 2017).

Following standard protocol, we also treat **CIFAR-10** as ID and **CIFAR-100** as near-OOD (semantically related) to assess detectors under tighter distributional shifts. Near-OOD is challenging because samples can lie close to the ID support and be mistaken as ID. We report the results in table.

Finally, we made an even more challenging near-OOD experiment by taking **Imagenette** Howard (2019) (a subset of ten classes from **ImageNet** Russakovsky et al. (2015)) as ID and, for each class in it, we selected from ImageNet a corresponding near-OOD class, i.e., semantically close to it (in Supp. we detail which specific classes from ImageNet we selected and why/how).

## 4.3 RESULTS

We report (i) **FPR95**: false positive rate of OOD samples at 95% true positive rate on ID; (ii) **AUROC**: area under the ROC curve. For all tables: best in **bold**, second best underlined.

**Fully unsupervised case**    The basic and standard fully unsupervised 'pure machine learning' baselines we used to compare and ablate our method are IF and $k-$NN on the raw pixel-space data. And then both of these methods again but now on the latent space of a standard AE and VAE; in addition, we also run a MSE method in the latter cases. We tried other standard methods too, but decided to limit the presentation only to these, since they were always the best performing and more consistent.

See Table 1 for the results. $k$-NN (pixel-space) outperforms AE+MSE; AE+$k$-NN (latent) barely improves on this, while our Comp.VAE+$k$-NN (latent) yields a clear gain, also over the vMF version.

**Semi-supervised case**    We can see in Tables 2 33 that our approach offers a systematic and consistent lowering of the FPR95, despite not achieving some of the best AUCs in the far OOD case, in both far and near OOD types. In the case of CIFAR-10 (ID) vs CIFAR-100 the gain is considerable and the AUC is comparable to the state-of-the-art results, while in the far OOD case it also is noticeable (in the cases without contrastive learning, while the methods using contrastive learning beat our result, but not by as much as w.r.t. the other methods that do not use contrastive learning).

In Table 3, the most challenging experiment for near OOD, full compression of all the hyperspherical coordinates beats the case of compressing only the first one (vMF method) in both FPR95 and AUC. This is to be expected if our hypothesis is correct, since the compression of more hyperspherical coordinates helps to reduce the sparsity and volume of the HD latent space even faster, as we discuss

Table 1: AUROC ($\uparrow$) and FPR95 ($\downarrow$) for anomaly detection methods on two datasets (all experiments run by us). Best in **bold**, second best underlined.

| AD Method | Mars Rover Mastcam | | Galaxy Zoo | |
|---|---|---|---|---|
| | AUROC $\uparrow$ | FPR95 $\downarrow$ | AUROC $\uparrow$ | FPR95 $\downarrow$ |
| $k$NN (pixel space) | 0.669 | 0.63 | 0.740 | 0.80 |
| Isolation Forest (pixel space) | 0.541 | 0.97 | 0.661 | 0.93 |
| AE + $k$NN (latent) | 0.681 | 0.64 | 0.754 | **0.72** |
| AE + IF (latent) | 0.591 | 0.93 | 0.712 | 0.82 |
| AE + MSE | 0.617 | 0.90 | 0.709 | 0.81 |
| VAE + $k$NN (latent) | 0.664 | 0.66 | 0.741 | 0.77 |
| VAE + IF (latent) | 0.530 | 0.94 | 0.700 | 0.84 |
| VAE + MSE | 0.653 | 0.88 | 0.730 | 0.80 |
| Comp.VAE (vMF) + $k$NN (latent) | 0.712 | 0.63 | 0.773 | 0.82 |
| *Comp.VAE* + $k$NN (latent) (ours) | **0.764** | **0.62** | **0.789** | 0.80 |

in more detail in Supp. Thus, the superior results of the similar, vMF-based method CIDER w.r.t. to ours in the far OOD case seems to be mostly caused by the enhancement of the method by the use of the contrastive learning techniques. We were not able to reproduce the reported results with our KNN* implementation. This is likely due to the fact that the mentioned references use other enhancement techniques (besides the contrastive learning) or architectural details. In any case, the backbone we use was the same for our Comp.VAE, which, thus, manages to get close to the state-of-the-art AUC results and give a state-of-the-art FPR95, even after being built on top of such a low-performing baseline (KNN*).

| Method | SVHN | | Places365 | | LSUN | | iSUN | | Texture | | Average | |
|---|---|---|---|---|---|---|---|---|---|---|---|---|
| | FPR$\downarrow$ | AUROC$\uparrow$ | FPR$\downarrow$ | AUROC$\uparrow$ | FPR$\downarrow$ | AUROC$\uparrow$ | FPR$\downarrow$ | AUROC$\uparrow$ | FPR$\downarrow$ | AUROC$\uparrow$ | FPR$\downarrow$ | AUROC$\uparrow$ |
| *Without Contrastive Learning* | | | | | | | | | | | | |
| MSP | 59.66 | 91.25 | 62.46 | 88.64 | 45.21 | 93.80 | 54.57 | 92.12 | 66.45 | 88.50 | 57.67 | 90.86 |
| Energy | 54.41 | 91.22 | 42.77 | 91.02 | 10.19 | 98.05 | 27.52 | 95.59 | 55.23 | 89.37 | 38.02 | 93.05 |
| ODIN | 53.78 | 91.30 | 43.40 | 90.98 | 10.93 | 97.93 | 28.44 | 95.51 | 55.59 | 89.47 | 38.43 | 93.04 |
| GODIN | 18.72 | 96.10 | 55.25 | 85.50 | 11.52 | 97.12 | 30.02 | 94.02 | 33.58 | 92.20 | 29.82 | 92.97 |
| Mahalanobis | 9.24 | 97.80 | 83.50 | 69.56 | 67.73 | 73.61 | 6.02 | 98.63 | 23.21 | 92.91 | 37.94 | 86.50 |
| KNN | 27.97 | 95.48 | 18.50 | 96.84 | 24.68 | 95.52 | 26.74 | 94.96 | 47.84 | 89.93 | 29.15 | **94.55** |
| KNN* | 91.0 | 66.4 | 90.0 | 59.6 | 90.0 | 59.9 | 91.6 | 57.9 | 88.9 | 55.7 | 90.3 | 59.9 |
| Comp.VAE (vMF) | 51.8 | 81.3 | 52.6 | 79.7 | 46.8 | 81.5 | 50.2 | 80.6 | 72.3 | 76.1 | 54.7 | 79.8 |
| Comp.VAE (*ours*) | 15.6 | 93.6 | 20.4 | 91.3 | 15.6 | 93.9 | 17.1 | 93.1 | 32.2 | 86.9 | **20.2** | 91.8 |
| *With Contrastive Learning* | | | | | | | | | | | | |
| CE + SimCLR | 6.98 | 99.22 | 54.39 | 86.70 | 64.53 | 85.60 | 59.62 | 86.78 | 16.77 | 96.56 | 40.46 | 90.97 |
| CSI | 37.38 | 94.69 | 38.31 | 93.04 | 10.63 | 97.93 | 10.36 | 98.01 | 28.85 | 94.87 | 25.11 | 95.71 |
| SSD+ | 2.47 | 99.51 | 22.05 | 95.57 | 10.56 | 97.83 | 28.44 | 95.67 | 9.27 | 98.35 | 14.56 | 97.38 |
| ProxyAnchor | 39.27 | 94.55 | 43.46 | 92.06 | 21.04 | 97.02 | 23.53 | 96.56 | 42.70 | 93.16 | 34.00 | 94.67 |
| KNN+ | 2.70 | 99.61 | 23.05 | 94.88 | 7.89 | 98.01 | 24.56 | 96.21 | 10.11 | 97.43 | 13.66 | 97.22 |
| CIDER | 2.89 | 99.72 | 23.88 | 94.09 | 5.45 | 99.01 | 20.21 | 96.64 | 12.33 | 96.85 | 12.95 | 97.26 |

Table 2: Results on CIFAR-10 as ID dataset for Far-OOD (results for the other methods taken from Ming et al. (2023).

**3d visualization of the hypersphere** Our method of volume compression allows for a direct $3-$dimensional visualization of the HD latent space. It shows the latent of a standard VAE is uniformly distributed over the sphere and not informative, contrary to our case (Figure 1). This was done by averaging the 256 latent dimensions into three (first 85, second 85, and the remaining 86), and normalizing each of the resulting 3D vectors to the sphere. Each HD latent vector could thus be plotted as a point in 3D and in this way visualize the HD latent space in a rather direct way. Furthermore, the $3-$dimensional visualization shows something remarkable in the case of the compressed VAE: the latent samples are compressed towards a small 'island' on the hypersphere and away from the equator, but the classes are actually *visible*, unlike the case of the standard VAE. We believe that it is because the samples are now located away from the equator, in a region with a much lower volume, where there are many fewer possibilities to realize this clustering in terms of different possible latent point configurations.

Table 3: Near-OOD results. Left: CIFAR-10 (ID) vs CIFAR-100 (results for the other methods taken from Ghosal et al. (2024). Right: Imagenette (ID) vs close ImageNet classes (all run by us).

| Methods | Near-OOD | | Methods | Near-OOD | |
|---|---|---|---|---|---|
| | FPR95↓ | AUROC↑ | | FPR95↓ | AUROC↑ |
| MSP | 64.66 | 85.28 | KNN* | 89.9 | 58.9 |
| ODIN | 52.32 | 88.90 | Comp.VAE (vMF) | 82.3 | 66.4 |
| GODIN | 60.69 | 82.37 | | | |
| Energy score | 58.66 | 86.06 | Comp.VAE (*ours*) | **78.8** | **68.3** |
| ReAct | 53.51 | 88.96 | | | |
| GradNorm | 65.44 | 79.31 | | | |
| LogitNorm | 55.08 | 88.03 | | | |
| DICE | 58.60 | 87.11 | | | |
| Mahalanobis | 87.71 | 78.93 | | | |
| KNN | 58.34 | 87.90 | | | |
| SNN | 50.10 | **89.80** | | | |
| KNN* | 90.0 | 61.5 | | | |
| Comp.VAE (vMF) | 61.1 | 77.4 | | | |
| Comp.VAE (ours) | **23.2** | 89.5 | | | |

## 5 DISCUSSION AND CONCLUSION

We propose to convert the latent variables of a VAE to hyperspherical coordinates. This allows moving latent vectors on a small island of the hypersphere. We showed that this modification improves AD (in both fully unsupervised and semi-supervised modalities, for the latter also in the far and near OOD types) as our results outperform other comparable methods in many cases.

We report state-of-the art results on the FPR95 metric in the semi-supervised experiments of CIFAR-10 (ID) vs far OOD standard benchmarks (w.r.t. methods that, like ours, do not use contrastive learning techniques, which by themselves seem to enhance most methods regardless of their inner details), and a very strong one in the case of near OOD for CIFAR-10 (ID) vs CIFAR-100. We also explored the complex and difficult ImageNet-based near OOD scenario of Imagenette (ID) vs close ImageNet classes. Our method showed the best results in both metrics, while at the same time providing an important ablation w.r.t. another close type of methods based on vMF distributions, which our own method can actually reproduce as a sub-case (compress only one angular coordinate instead of all of them). Our compression method can be used in *most of the* VAE variations, since it just affects how the KLD term is computed.

The transformation from Cartesian to hyperspherical coordinates adds processing time, despite a vectorized implementation. Computation for an epoch was 32% more expensive in time using 200 latent dimensions. As the number of dimensions grows, the computation becomes more expensive (see Supp. for details).

The constants $\alpha_{i,j}$, $\beta_{i,j}$ multiplying the elements of the hyperspherical loss are proportional to $1/\sqrt{k+1}$, where $k$ is the coordinate index. The number of free hyperparameters is thus reduced to *four scalars* for the angular losses and four scalars for the radial ones. However, we found that only two of them ($\beta$ and $\alpha_{\mu,r}$, the latter only occasionally) need adjustment when changing datasets (at least for the ones used in this paper). Those parameters can be found in the code provided here (to be updated upon acceptance).

This work stemmed from the hypothesis that the latent of a VAE is very sparse, which limits its ability to be used as a generative model because most of the latent is not sampled during training. Recent work showed that compressing the latent using hyperspectral coordinates does indeed improve the generation of new data when sampling the prior Ascárate et al. (2025). In this new work we show that, as expected, reducing latent sparsity also helps to detect anomalies. We speculate that controlling and reducing the sparsity of the high dimensional latent manifold should be beneficial for other tasks such as classification, which we aim to explore next.

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
