# Supplementary material for "VAE with Hyperspherical Coordinates: Improving Anomaly Detection From Hypervolume-Compressed Latent Space"

Anonymous

September 25, 2025

# Contents

# 1 Hyperspherical coordinates

## 1.1 Conversion between Cartesian and hyperspherical coordinates

For reference, we provide the standard formulas for converting between Cartesian and spherical coordinates.

In $n$ dimensions, given a set of Cartesian coordinates $x_k$ with $k \in \{1, \ldots, n\}$, the hyperspherical coordinates are defined by a radius $r$ and $n-1$ angles $\varphi_k$ with $k \in \{1, \ldots, n-1\}$; $\varphi_k \in [0, \ldots, \pi]$ for $k \in \{1, \ldots, n-2\}$ and $\varphi_{n-1} \in [0, \ldots, 2\pi)$.

From hyperspherical to Cartesian conversion:

$$
\begin{aligned}
x_1 &= r\cos(\varphi_1) \\
x_2 &= r\sin(\varphi_1)\cos(\varphi_2) \\
x_2 &= r\sin(\varphi_1)\sin(\varphi_2)\cos(\varphi_3) \\
&\vdots \\
x_{n-1} &= r\sin(\varphi_1)\sin(\varphi_2)\ldots\sin(\varphi_{n-2})\cos(\varphi_{n-1}) \\
x_n &= r\sin(\varphi_1)\sin(\varphi_2)\ldots\sin(\varphi_{n-2})\sin(\varphi_{n-1})
\end{aligned}
\tag{1}
$$

From Cartesian to hyperspherical conversion:

$$
\begin{aligned}
r &= \sqrt{x_n^2 + x_{n-1}^2 + \ldots + x_2^2 + x_1^2} \\
\cos(\varphi_1) &= \frac{x_1}{\sqrt{x_n^2 + x_{n-1}^2 + \ldots + x_2^2 + x_1^2}} \\
\cos(\varphi_2) &= \frac{x_2}{\sqrt{x_n^2 + x_{n-1}^2 + \ldots + x_2^2}} \\
&\vdots \\
\cos(\varphi_{n-2}) &= \frac{x_{n-2}}{\sqrt{x_n^2 + x_{n-1}^2 + x_{n-2}^2}} \\
\cos(\varphi_{n-1}) &= \frac{x_{n-1}}{\sqrt{x_n^2 + x_{n-1}^2}}
\end{aligned}
\tag{2}
$$

## 1.2 Vectorized code for converting between Cartesian and hyperspherical coordinates

This code is accessible here and provided below for reference. It uses vectorized operations that are differentiable by torch. The correct vectorization of this script was key to make our proposal feasible in practice, since the introduction of any Python **for loop** increased the processing time considerably.

```python
import torch

def cart_to_cos_sph (x, device):
    m = x.size(0)
    n = x.size(1)
    mask = torch.triu(torch.ones(n, n)).to(device)
    mask = torch.unsqueeze(mask, dim=0)
```

```
mask = mask.expand(m, n, n)
X = torch.unsqueeze(x, dim=1).expand(m, n, n)
X_squared = torch.square(X)
X_squared_masked = X_squared * mask
denom = torch.sqrt(torch.sum(X_squared_masked, dim=2)+0.001)
cos_phi = x / denom
return cos_phi[:, 0:n-1]
```

## 2    Concentration of measure effects

### 2.1    Concentration of Measure: basic idea

In this appendix, we collect the results of simple experiments that clearly show the concentration of measure effects that occur in high dimensions. In Fig. 1a), we show the distribution of a simple Normal distribution in 2 dimensions (left), and the histogram for the norm of the samples (right). In b), the same but for a Normal distribution in 100 dimensions. In Fig. 2a), we show the histogram for the angle between two random samples from a Normal distribution in 2 dimensions (left), and the same but for a Normal distribution in 100 dimensions (right). In b), we display a schematic diagram of the mass concentration of the uniform measure of the hypersphere in very high dimensions. The intuition in this diagram comes from the more precise result [1] which states that, for **any** given $y \in \mathbb{R}^n$, if we define on the hypersphere an '**equatorial**' slice of width $\epsilon > 0$ as $T_y(\epsilon) \doteq \left\{ z \in \mathbb{S}^{n-1} / \mid (z,y) \mid \leq \epsilon/2 \right\}$, then its volume according to the uniform measure satisfies the following concentration inequality:

$$\mathbb{P}\left[T_y(\epsilon)\right] \geq 1 - \sqrt{2\pi} \exp(-\frac{n\epsilon^2}{2}). \tag{3}$$

The previous inequality shows that, in very high dimensions, the equatorial slice $T_y(\epsilon)$ occupies a huge portion of the total volume, even for a very small width.

Finally, with this in place, we can understand the peculiar shape that a high dimensional Normal distribution takes when expressed in hyperspherical coordinates (Fig.3).

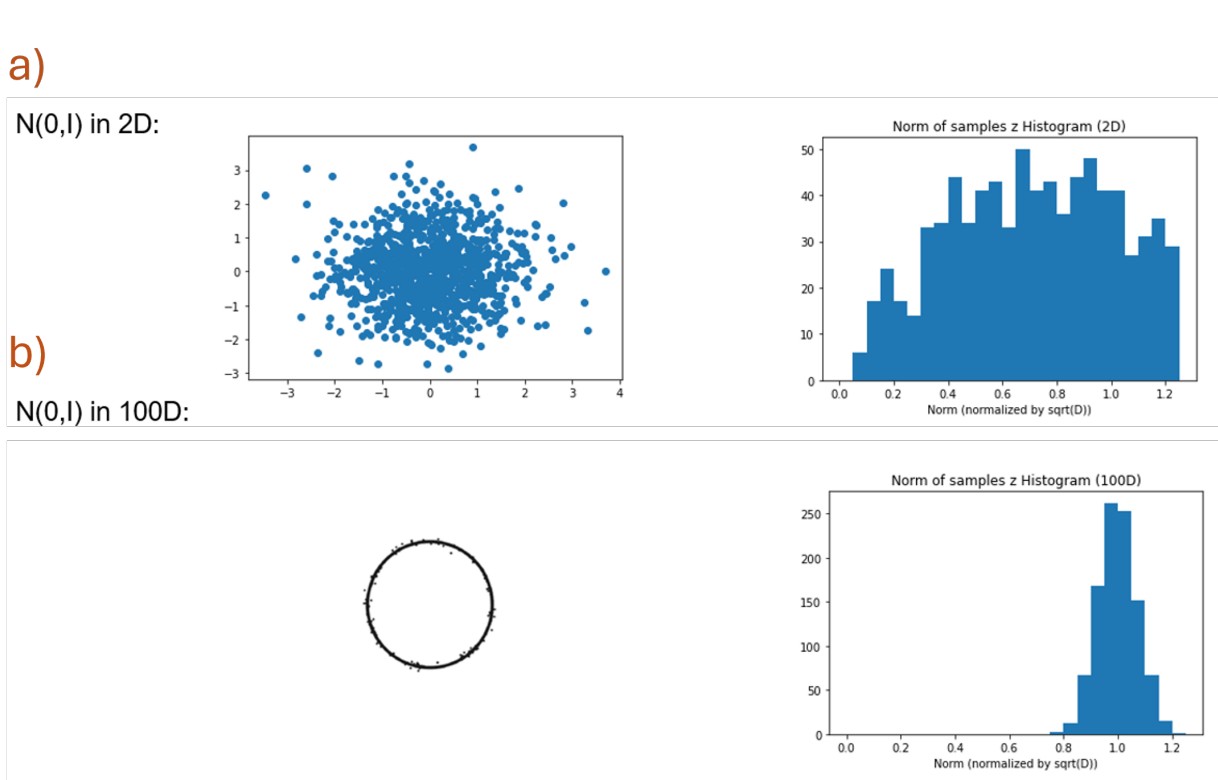

**Figure 1:** Measure concentration, norm (left image in b), adapted from [2])

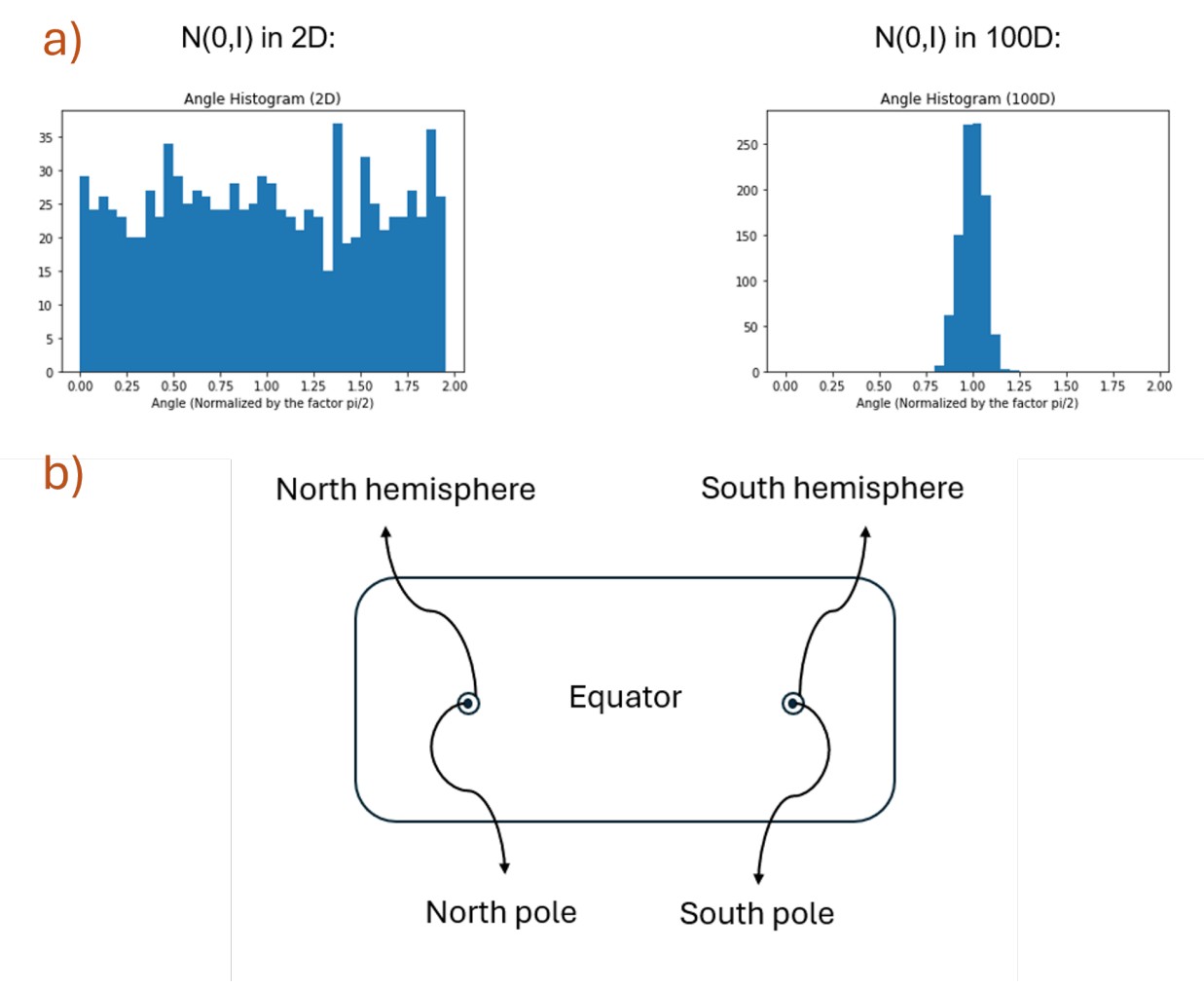

**Figure 2:** a) Measure concentration, angle; b) Schematic diagram of the mass concentration of the uniform measure of the hypersphere in very high dimensions: most of the volume is in any equator.

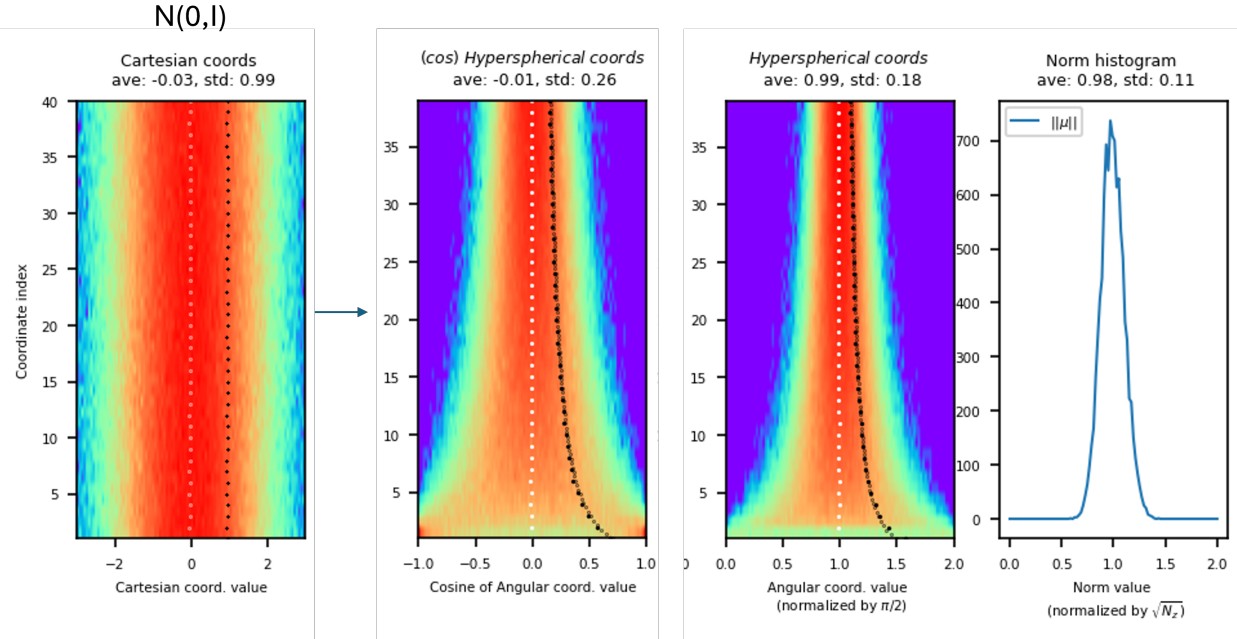

**Figure 3:** High dimensional Normal distribution in hyperspherical coordinates. For the first three images from the left, each horizontal slice at some vertical index value shows the color coded histogram (red, high density; blue, low density) for the range of the coordinate of that index; the vertical axis stacks all the histograms for all the dimensions (in this example, 40). The white dots represent the mean and the black dots represent the standard deviation of the corresponding histogram. The numbers on top are the total mean and standard deviation of all these previous values taken together.

## 2.2 Hypervolume reduction in hyperspherical coordinates

The hypervolume element of the hypersphere $\mathbb{S}_R^{n-1}$ is given by the following expression when using hyperspherical coordinates[1]:

$$\mathrm{d}V_{\mathbb{S}_R^{n-1}} = R^{n-1} \sin^{n-2}(\varphi_1) \sin^{n-3}(\varphi_2) \cdots \sin(\varphi_{n-2}) \mathrm{d}\varphi_1 \mathrm{d}\varphi_2 \cdots \mathrm{d}\varphi_{n-1} \tag{4}$$

The volume can be reduced much faster and effectively by reducing the angular coordinates (away from the equators), than by either reducing just the radius of the hypersphere or, equivalently, all of the Cartesian coordinates. The higher the dimension, the more pronounced this difference becomes because each added dimension $k$ adds extra powers of $\sin \varphi_k$ in the hypervolume element (see section Hypervolume reduction in hyperspherical coordinates). The further the angles from $\pi/2$, the smaller the infinitesimal hypervolume element becomes as it is multiplied by an increasingly smaller quantity lower than 1. This is a purely geometric effect.

As mentioned, the hypervolume element of the hypersphere $\mathbb{S}_R^{n-1}$ is given by the expression of equation (4) when using hyperspherical coordinates. In the small angle regime, where $\sin \varphi \approx \varphi$, we can approximately integrate this expression for an angular coordinate hypercube $[0, \varphi_0]^{n-1}$, and the result is proportional to $v_0 = R^{n-1} \varphi_0^{n(n-1)/2}$. If now we reduce the size of the angular coordinate hypercube by a schedule of the form $\varphi_t = \varphi_0(1-t)$, $t \in [0, 1]$, then we can compare the percentage of hypervolume being reduced from the initial value, while keeping $R$ fixed, to the percentage obtained by reducing the size of the hypersphere by an schedule of the form $R_t = R(1-t)$, $t \in [0, 1]$, while keeping $\varphi_0$ fixed (this second case is equivalent to reducing all the Cartesian coordinates at once, because $r^2 = x_n^2 + x_{n-1}^2 + \ldots + x_2^2 + x_1^2$). Indeed, we get, respectively, $v_t = v_0(1-t)^{n(n-1)/2}$ and $v_t = v_0(1-t)^{n-1}$.

In Fig.4 we plot the behavior of $v_t/v_0$ in terms of the reduction of the coordinate, given by $(1-t)$, for three, increasing values of dimension $n$. As we can see, already in dimension 20 (bottom figure in the panel), there is a sharp decrease in volume in the angular case as soon as one decreases the angular coordinates by a minimal amount; in comparison to the radial/Cartesian coordinate case, the abrupt decrease in volume looks almost discontinuous (in Statistical Physics, this type of behavior often occurs when the system undergoes a phase transition [3]).

Reducing the radius of the ball centered at the prior for the means $\mu$ given by the encoder is how the standard VAE tries to reach this prior: this corresponds to the term of the form $\| \mu \|^2$ in the KLD part in Cartesian coordinates, $\mathrm{KLD} = -\frac{1}{2} \sum_{k=1}^n \left(1 + \log(\sigma_k^2) - \sigma_k^2\right) + \frac{1}{2} \| \mu \|^2$ (see [4, 5]).

The most general form is $\| \mu - \mu_0 \|^2$, where $\mu_0$ is the prior value, the north pole in our case; that is, in both this case and in our method, the means $\mu$ given by the encoder are encouraged to go to the north pole by the losses, but what is really important in high dimensions is **how** we reach it, because of the peculiar 'equatorial' presentation of the volume in the hypersphere: we claim that our method using hyperspherical coordinates is better suited for this, since it allows to reduce the volume much faster by directly engaging with moving the samples away from the equators, thing which the radial reduction cannot do by construction.

### 2.2.1 Comparison with von Mises-Fisher-based approaches

The case when one reduces *only* a single angular coordinate, e.g., $\varphi_1$, is exactly like the radial one, since in equation (4) the exponent for this angle is $n-2$ which upon integration becomes $n-1$, like for the radial coordinate. Thus, the tendencies in the analogous of Figure 4 look exactly the same. This case corresponds to varying the single scalar variance parameter on a von Mises-Fisher distribution

---

[1](see https://en.wikipedia.org/wiki/N-sphere)

on the hypersphere, which is defined as an isotropic (hence the single free parameter rather than a vector or matrix for this variance) Gaussian whose domain is restricted to the hypersphere. This approach has been used recently in the literature when dealing with encodings into the hypersphere [6, 7], but as we mentioned, it cannot reduce the hypervolume as fast as our method in case one wishes to do so.

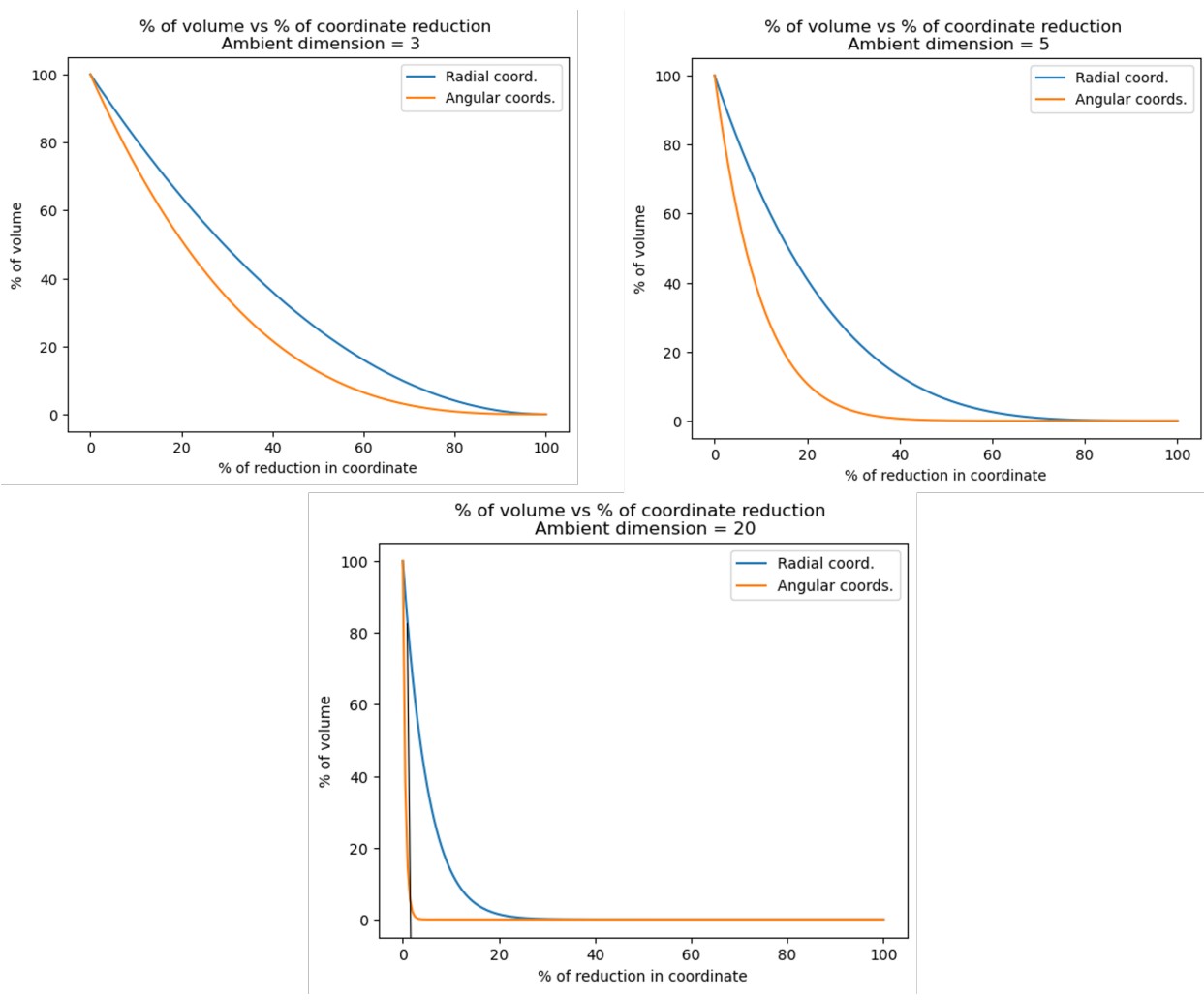

**Figure 4:** Hypervolume element reduction comparison: $(1 - t)^{n-1}$ vs. $(1 - t)^{n(n-1)/2}$.

# 3 Re-writing of the KLD term

In this appendix, we make explicit the steps to go from the standard form of the KLD term in the VAE to the one we used as a starting point, in equation (1) of the main paper, for our own KLD in hyperspherical coordinates.

In Cartesian coordinates, the KLD divergence between the estimated posterior defined by $\mu_k$ and $\sigma_k$ and the prior defined by $\mu_k^p$ and $\sigma_k^p$:

$$\text{KLD}_{\text{CartCoords}}^{w/Prior} = \frac{1}{2} \sum_{k=1}^{n} \left[ \left( \frac{\sigma_k}{\sigma_k^p} \right)^2 - \log \left( \frac{\sigma_k}{\sigma_k^p} \right)^2 - 1 + \frac{\left( \mu_k - \mu_k^p \right)^2}{\left( \sigma_k^p \right)^2} \right] \tag{5}$$

A Taylor approximation (up to second order) of the part for sigma around its prior yields for some constants $\gamma_k$ and $\widetilde{\gamma}_k$:

$$\text{KLD}_{\text{CartCoords}}^{w/Prior} \approx \sum_{k=1}^{n} \left[ \gamma_k \left( \sigma_k - \sigma_k^p \right)^2 + \widetilde{\gamma}_k \left( \mu_k - \mu_k^p \right)^2 \right] \tag{6}$$

In practice, the optimization is performed over mini batches of data (of size $N_b$), using the objective below:

$$\text{KLD}_{\text{CartCoords}}^{w/Prior} \approx \frac{1}{N_b} \sum_{l=1}^{N_b} \sum_{k=1}^{n} \left( \gamma_k \left( \sigma_{k,l} - \sigma_k^p \right)^2 + \widetilde{\gamma}_k \left( \mu_{k,l} - \mu_k^p \right)^2 \right) \tag{7}$$

If we denote the corresponding batch statistics as $\mathbb{E}_b$ and $\sigma_b$, then, by using the basic formula,

$$\mathbb{E}_b[X^2] = \mathbb{E}_b[X]^2 + \sigma_b[X]^2, \tag{8}$$

we can write this objective as (we omit the constants for ease of reading)

$$\text{KLD}_{\text{CartCoords}}^{w/Prior} \approx \sum_{k=1}^{n} \left( \left( \mathbb{E}_b[\sigma_k] - \sigma_k^p \right)^2 + \sigma_b[\sigma_k]^2 + \left( \mathbb{E}_b[\mu_k] - \mu_k^p \right)^2 + \sigma_b[\mu_k]^2 \right) \tag{9}$$

# 4 Datasets

## 4.1 Used in the unsupervised experiments

### 4.1.1 Mars Rover Mastcam

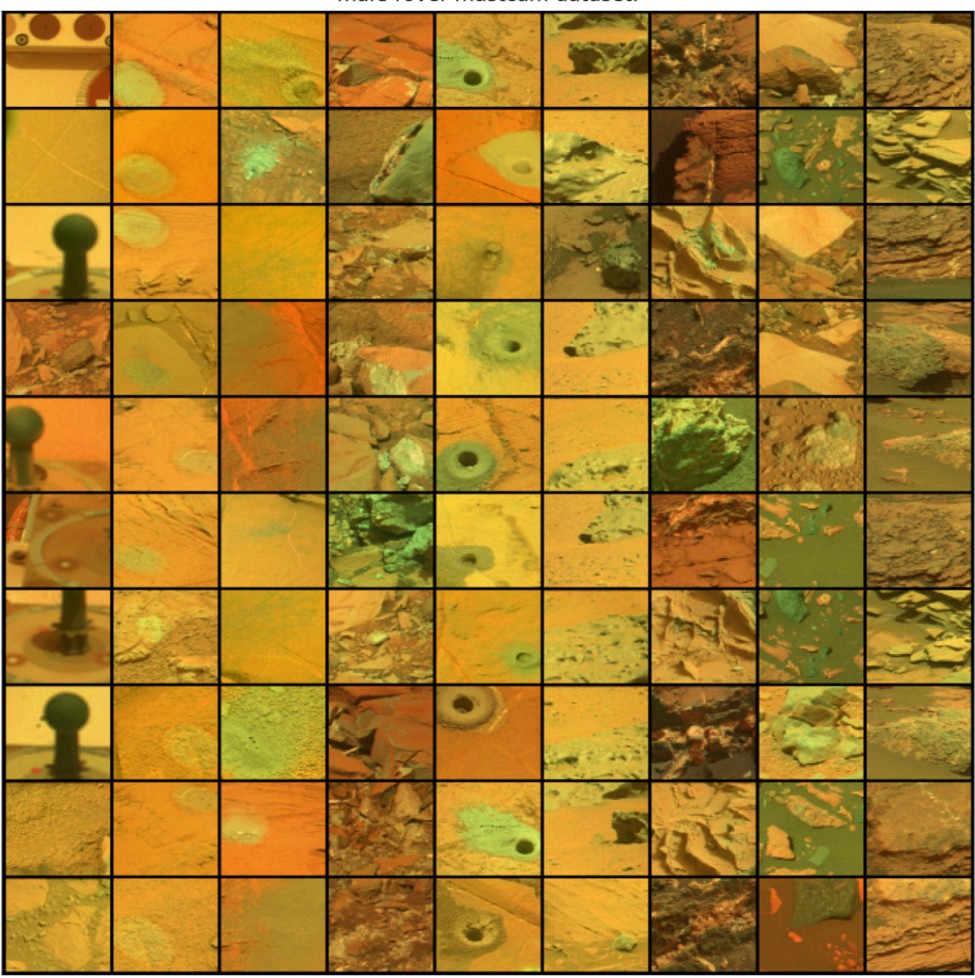

**Figure 5:** Examples from the test set of the Mars Rover Mastcam dataset. Images on the same column are in the same class.

## t-SNE: testing set

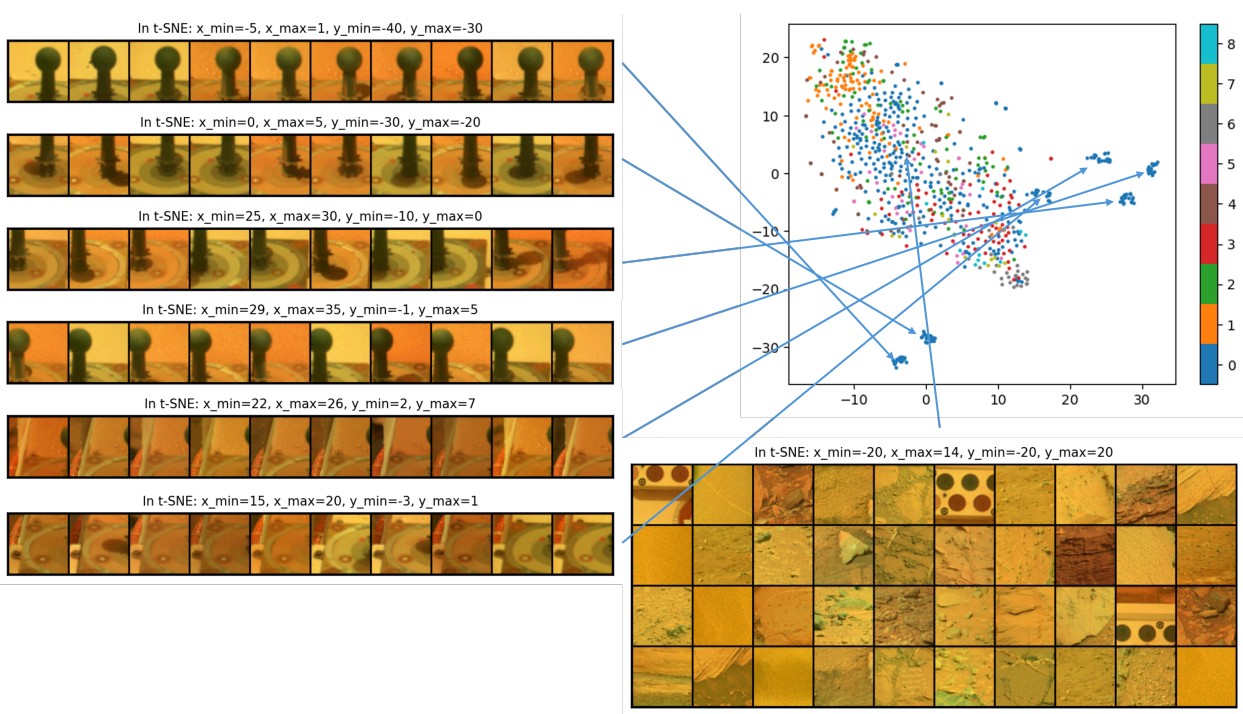

0:typical, 1:drt, 2:dump-pile, 3:broken-rock, 4:drill-hole, 5:meteorite, 6:veins, 7:float, 8:bedrock

**Figure 6:** t-SNE of the test set of the Mars Rover Mastcam dataset (pixel-space).

**Figure 7:** The different sub-classes of the normal class (blue) of the test set of the Mars Rover Mastcam dataset.

### 4.1.2 Galaxy Zoo

Training (typical), first two rows:    Testing (novel), all rows:

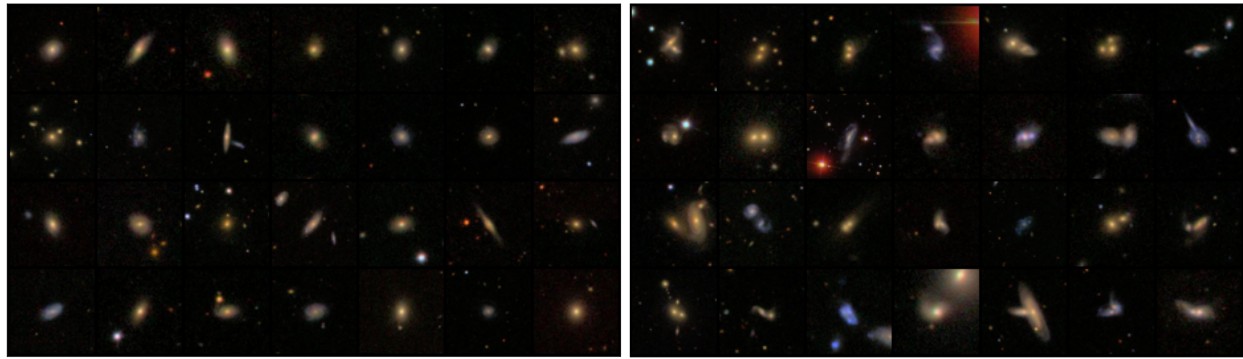

Testing (typical), last two rows .

**Figure 8:** Examples from the Galaxy Zoo dataset. Left: normal/typical. Right: novel.

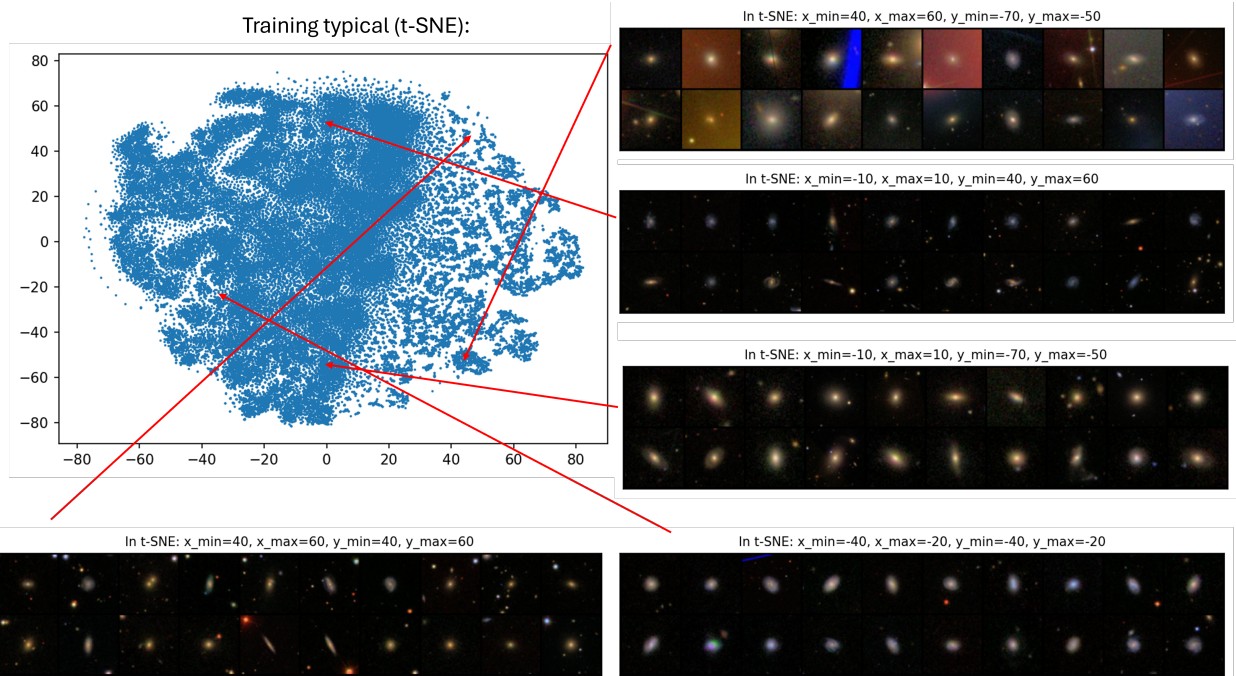

**Figure 9:** The different sub-classes of the normal class (blue) of the train set of the Galaxy Zoo dataset (t-SNE in pixel-space).

## 4.2  ImageNet-based dataset used in the semi-supervised experiments

The selection criterion for the near-OOD ImageNet classes was based on a simple semantic similarity search on the text corresponding to the classes' description.

**Table 1:** Imagenette → Near-OOD ImageNet mapping. The third column lists the WordNet/ILSVRC synset ID.

| Imagenette class | Near-OOD ImageNet class | Near-OOD synset ID |
| --- | --- | --- |
| tench | goldfish | n01443537 |
| English springer | Welsh springer spaniel | n02102177 |
| cassette player | CD player | n02988304 |
| chain saw | hand blower | n03483316 |
| church | mosque | n03788195 |
| French horn | cornet | n03110669 |
| garbage truck | moving van | n03796401 |
| gas pump | cash machine | n02977058 |
| golf ball | tennis ball | n04409515 |
| parachute | kite | n01608432 |

# 5 Results

Note: what we call the 'replica angle' (dashed lines) is the angle between the testing samples and the mean for all the normal test set (this angle value should give a rough idea about the angular size of the island). For more detailed information, we plot the full histograms of the hyperspherical coordinates.

## 5.1 Fully unsupervised experiments

### 5.1.1 Standard VAE on the Mars Rover Mastcam dataset

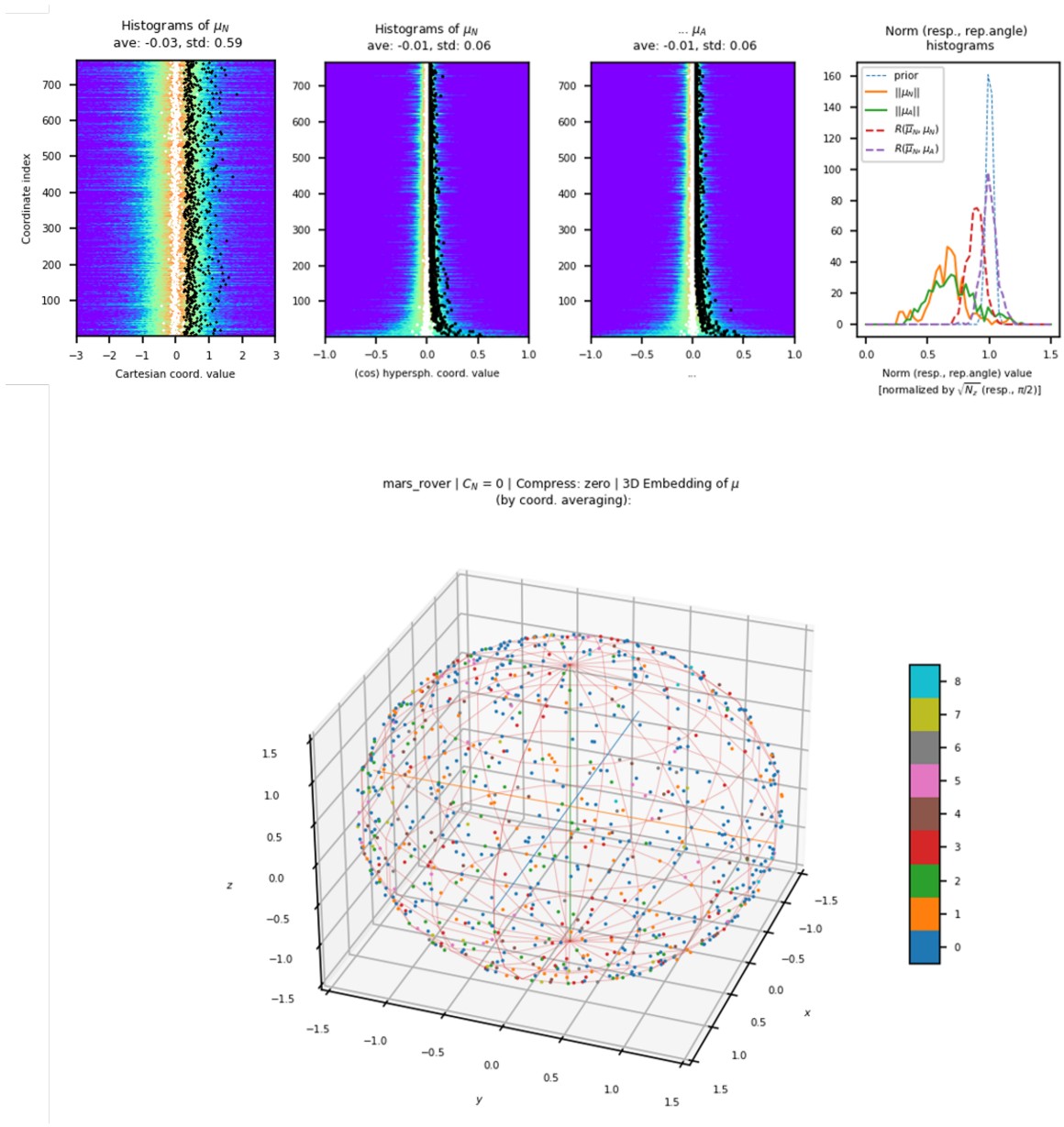

**Figure 10:** Results of the standard VAE training, on the test set of the Mars Rover Mastcam dataset. In the upper panel we show, from left to right, the histograms of $\mu$, using the same conventions as in Fig.3. The third histogram in this panel shows the norm histograms of $\mu$ and the replica angles.

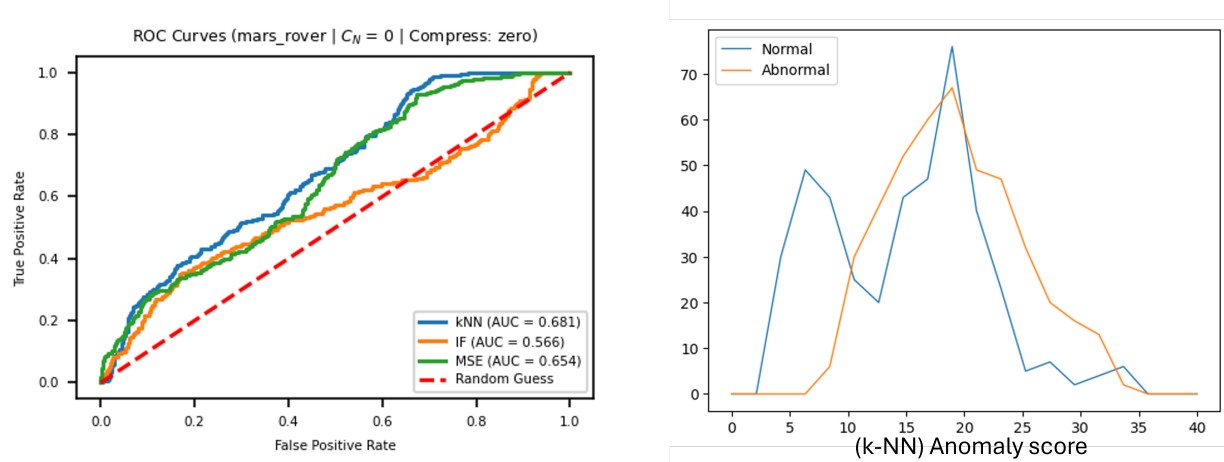

**Figure 11:** ROC curves and anomaly score histograms, results of the standard VAE training, on the test set of the Mars Rover Mastcam dataset. AUCs: Only values up to the third decimal place are reported, as results beyond that fluctuate across different runs of the same experiment.

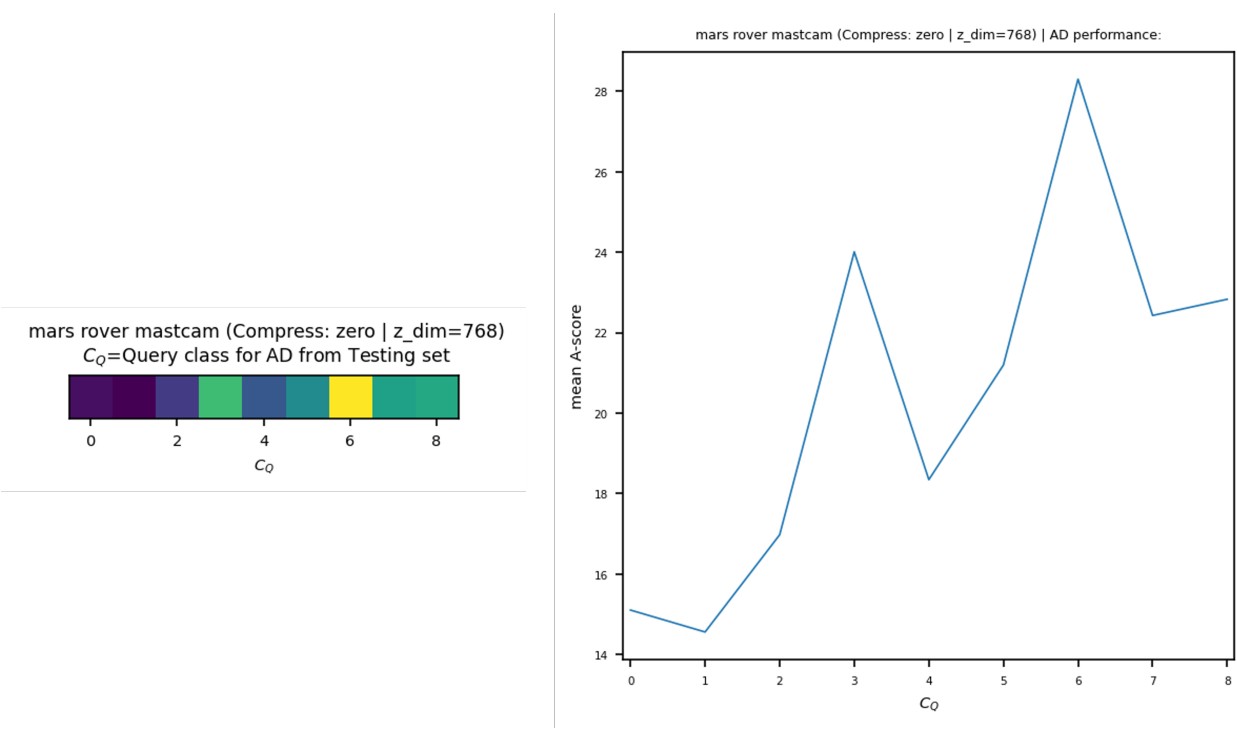

**Figure 12:** Mean $k-$NN anomaly score values for each class in the test set (0 is the typical/normal class), results of the standard VAE training, on the test set of the Mars Rover Mastcam dataset (blue means low, yellow is high).

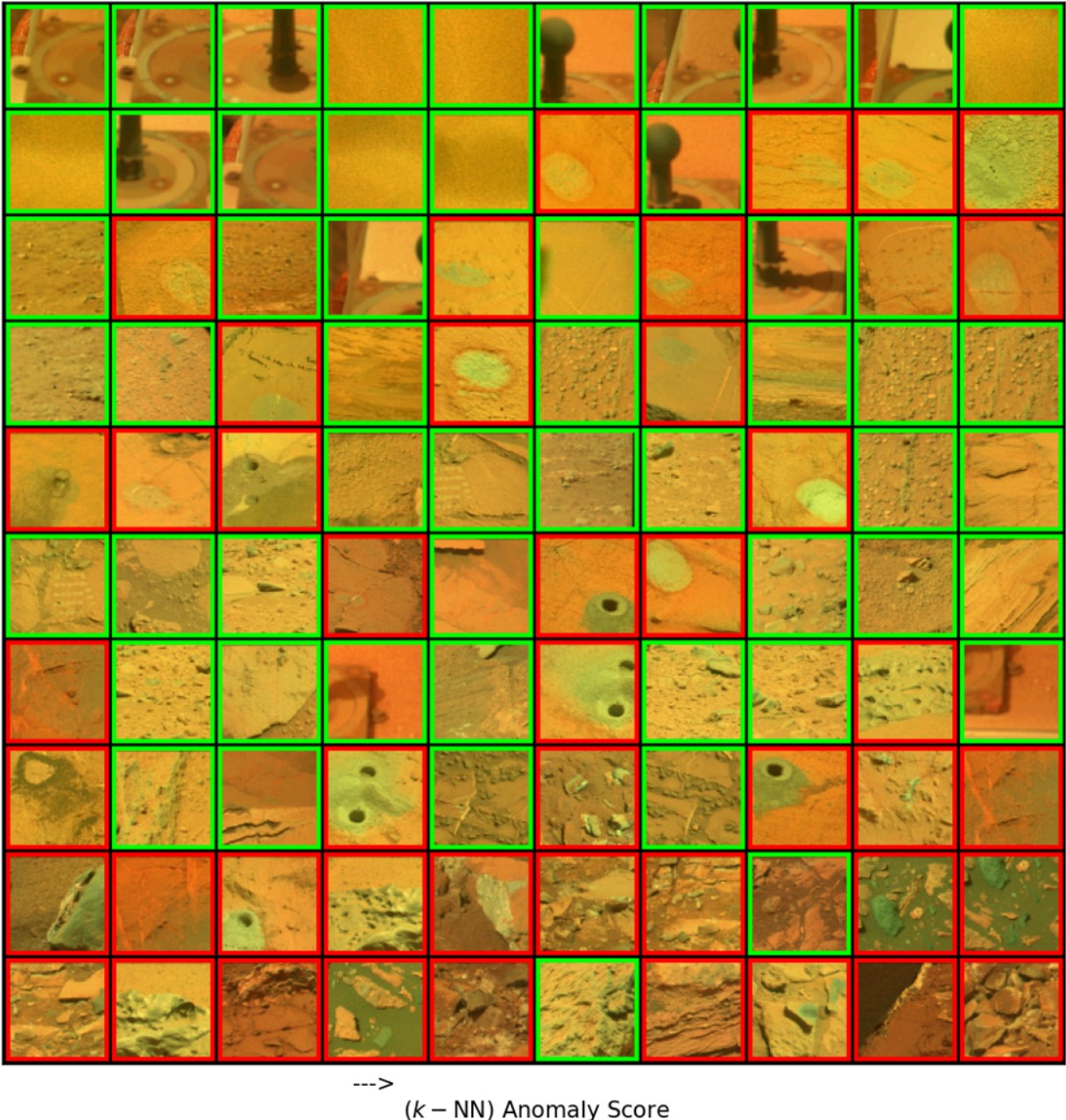

**Figure 13:** Results of the standard VAE training, on the test set of the Mars Rover Mastcam dataset. Showing the images from the test set but with the index sorted by the anomaly score (from low at the top to high at the bottom). Only displaying one every 8 images, starting from index 0. There are 420 typical images and 435 novel images in the testing set, for a total of 855. A perfect detection would show the top half of the total panel as normal (green framing) and the anomalies (red framing) at the bottom half.

mars rover mastcam (Compress: zero | z_dim=768) | AD performance:
(showing only Typical)

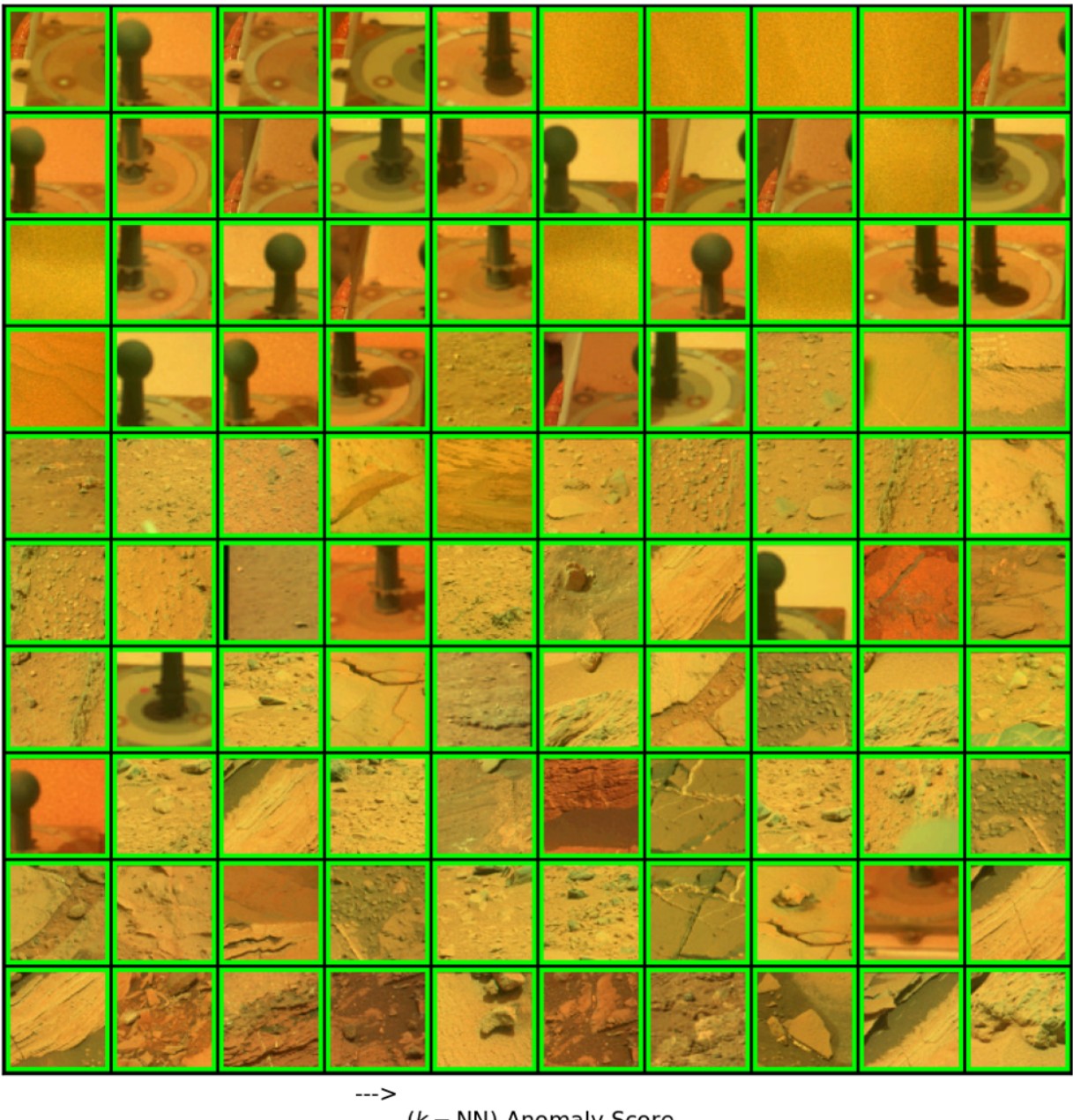

--->
$(k - NN)$ Anomaly Score
--->

**Figure 14:** Results of the standard VAE training, on the test set of the Mars Rover Mastcam dataset. Showing the images from the Typical test set only but with the index sorted by the anomaly score (from low to high). Only displaying one every 4 images, starting from index 0.

mars rover mastcam (Compress: zero | z_dim=768) | AD performance:
(showing only Novel)

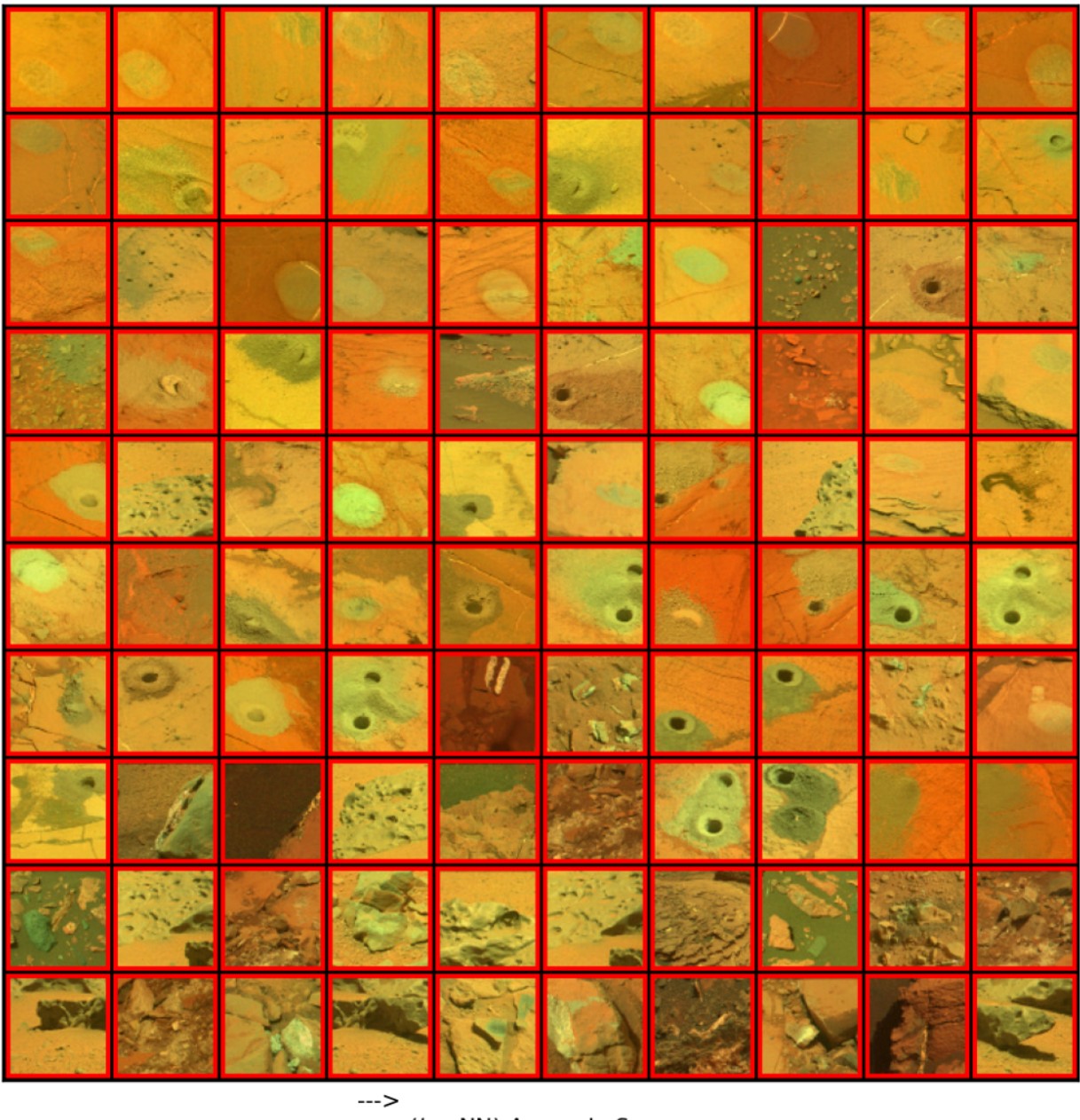

--->
$(k - NN)$ Anomaly Score
--->

**Figure 15:** Results of the standard VAE training, on the test set of the Mars Rover Mastcam dataset. Showing the images from the Novel test set only but with the index sorted by the anomaly score (from low to high). Only displaying one every 4 images, starting from index 0.

## 5.1.2 Comp.VAE on the Mars Rover Mastcam dataset

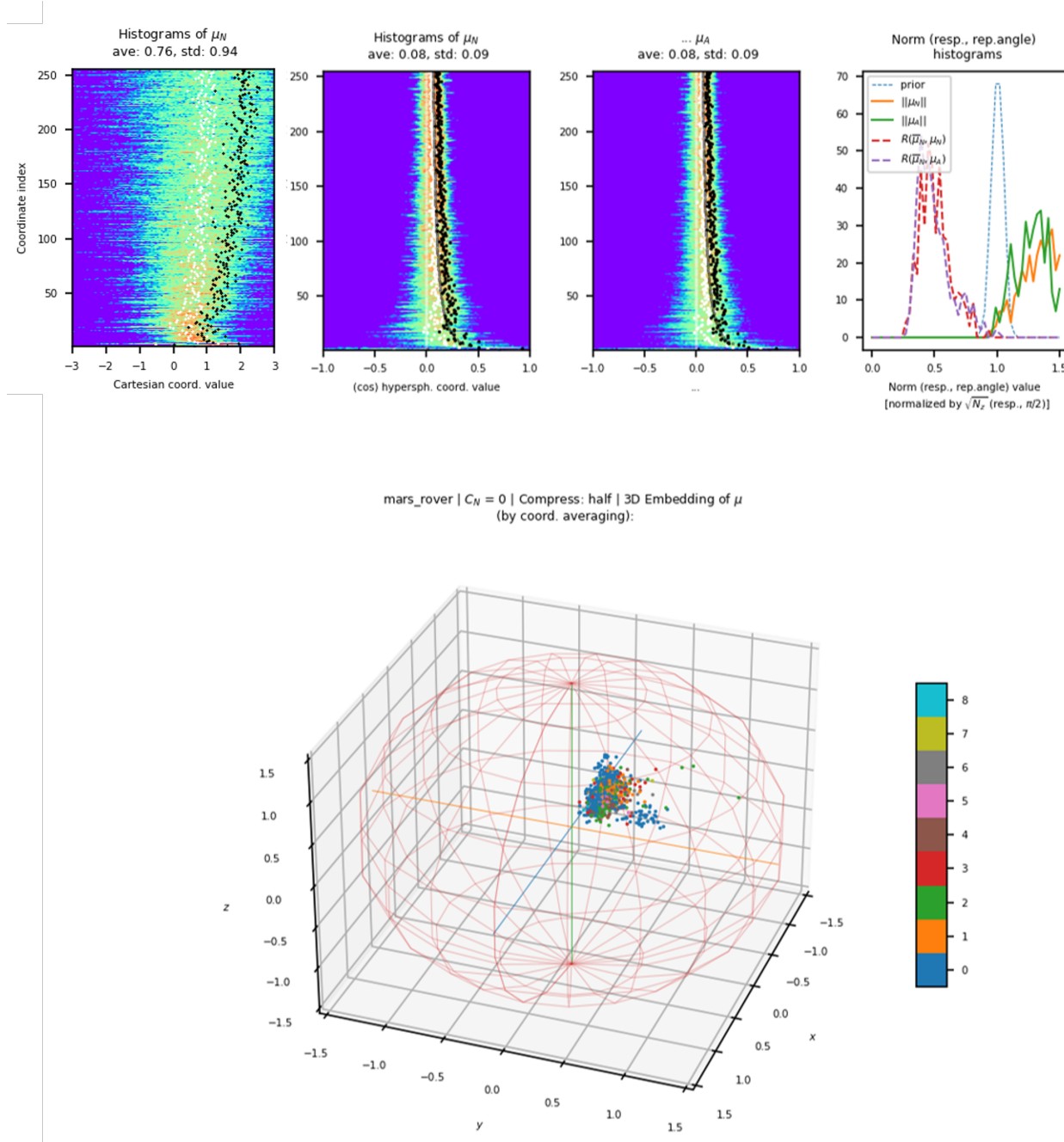

**Figure 16:** Results of the compressed VAE training, on the test set of the Mars Rover Mastcam dataset. Note how the means for the cosines of the hyperspherical angles are all shifted towards 1.0, but without collapsing the distribution. The compression direction in this case is given by the vector whose Cartesian coordinates are $(1, 1, ..., 1)$ (displayed as faint black dots in the cosine hyperspherical coordinates histograms for reference).

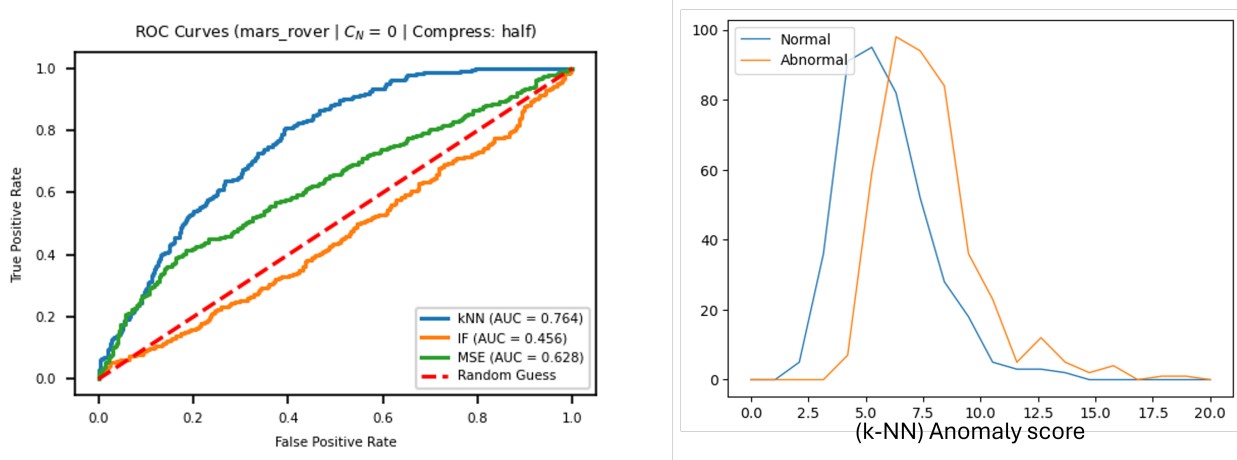

**Figure 17:** ROC curves and anomaly score histograms, results of the compressed VAE training, on the test set of the Mars Rover Mastcam dataset. Note the compression of the multiple peaks in the normal distribution of Figure 11 into a single one in the compressed version, and how the abnormal class is pushed more to the right.

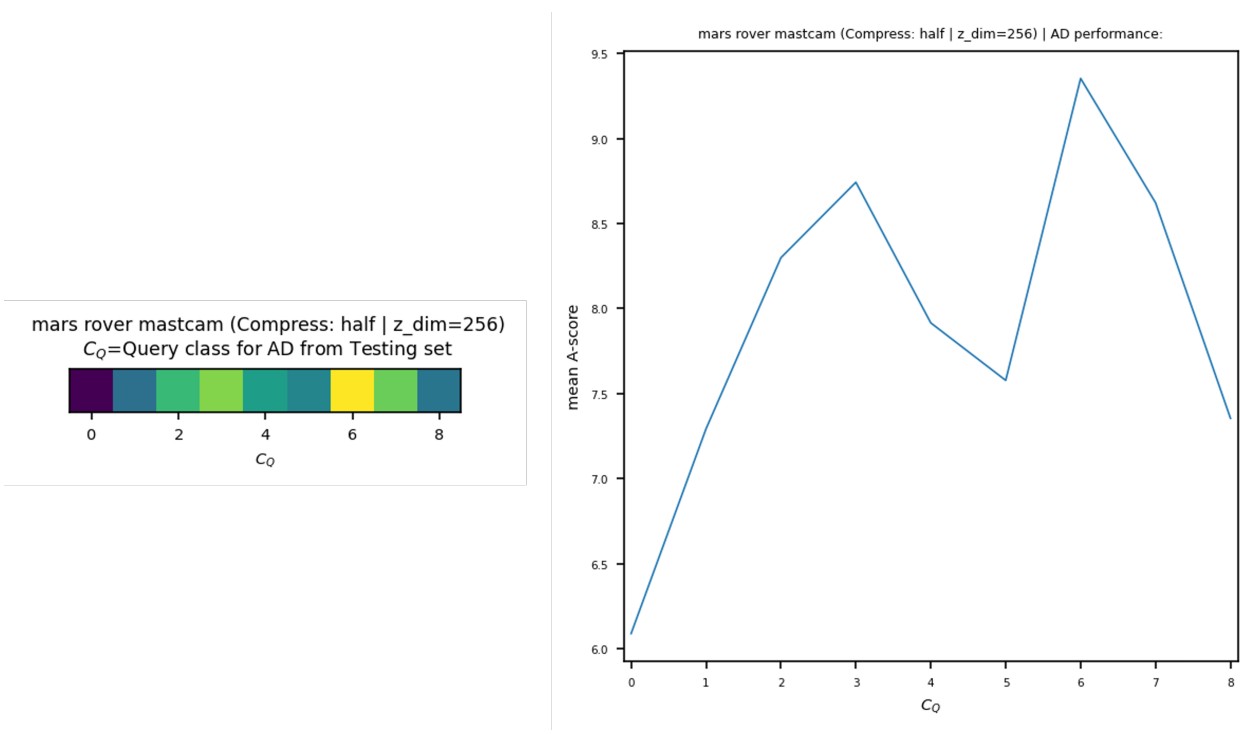

**Figure 18:** Mean $k-$NN anomaly score values for each class in the test set (0 is the typical/normal class), results of the compressed VAE training, on the test set of the Mars Rover Mastcam dataset.

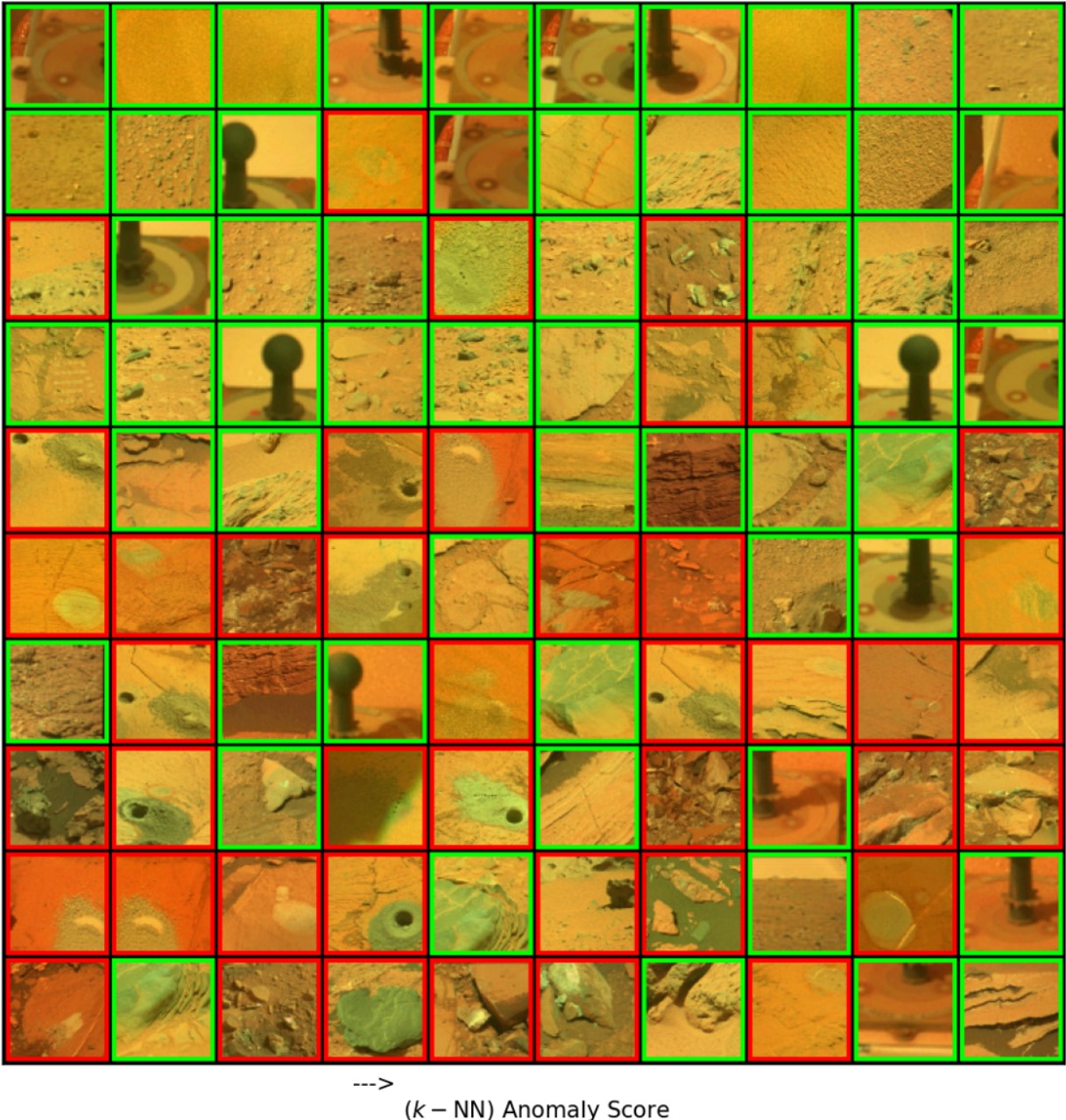

**Figure 19:** Results of the compressed VAE training, on the test set of the Mars Rover Mastcam dataset. Showing the images from the test set but with the index sorted by the anomaly score (from low at the top to high at the bottom). Only displaying one every 8 images, starting from index 0. There are 420 typical images and 435 novel images in the testing set, for a total of 855. A perfect detection would show the top half of the total panel as normal (green framing) and the anomalies (red framing) at the bottom half.

mars rover mastcam (Compress: half | z_dim=256) | AD performance:
(showing only Typical)

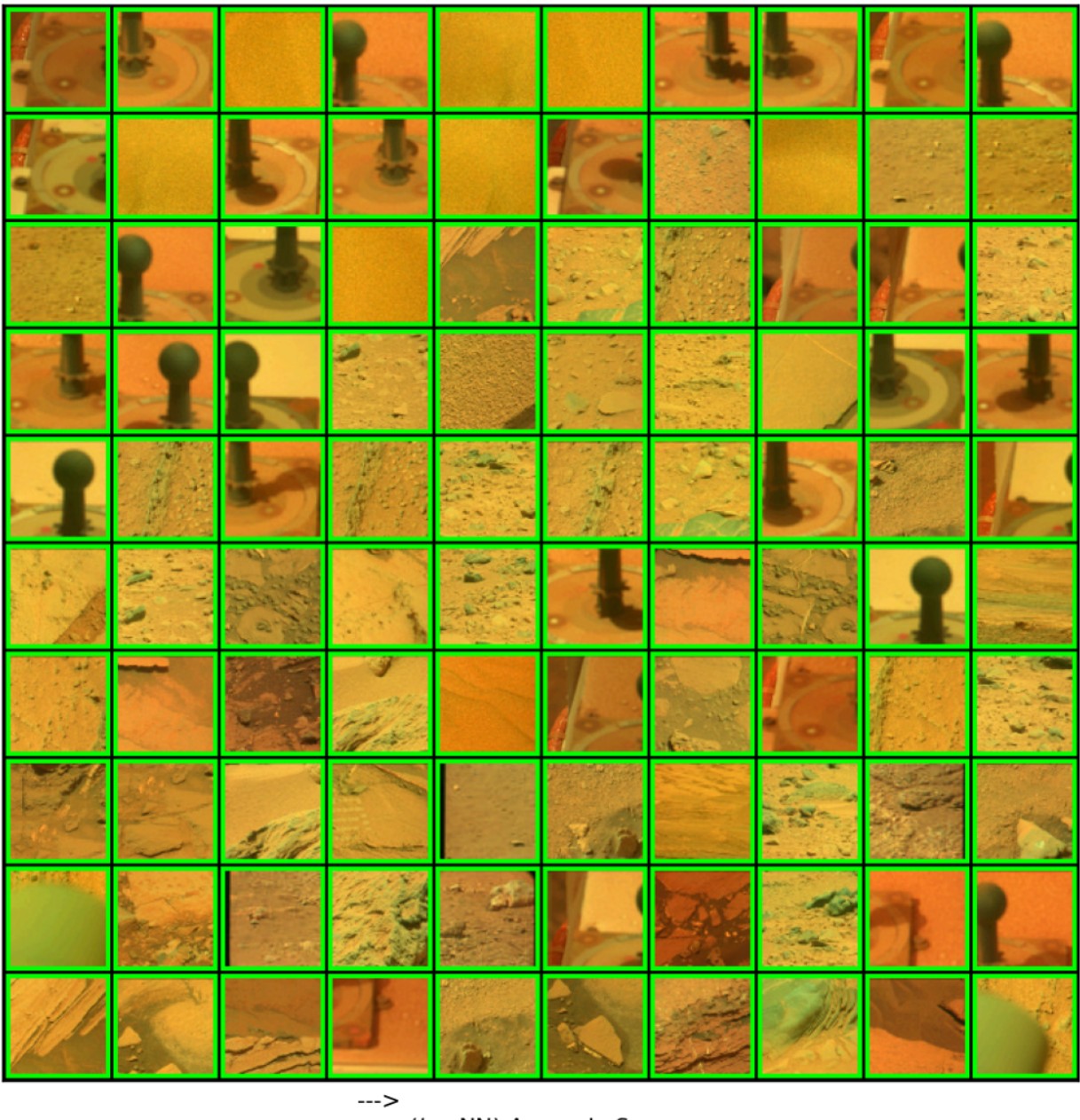

--->
$(k-NN)$ Anomaly Score
--->

**Figure 20:** Results of the compressed VAE training, on the test set of the Mars Rover Mastcam dataset. Showing the images from the Typical test set only but with the index sorted by the anomaly score (from low to high). Only displaying one every 4 images, starting from index 0.

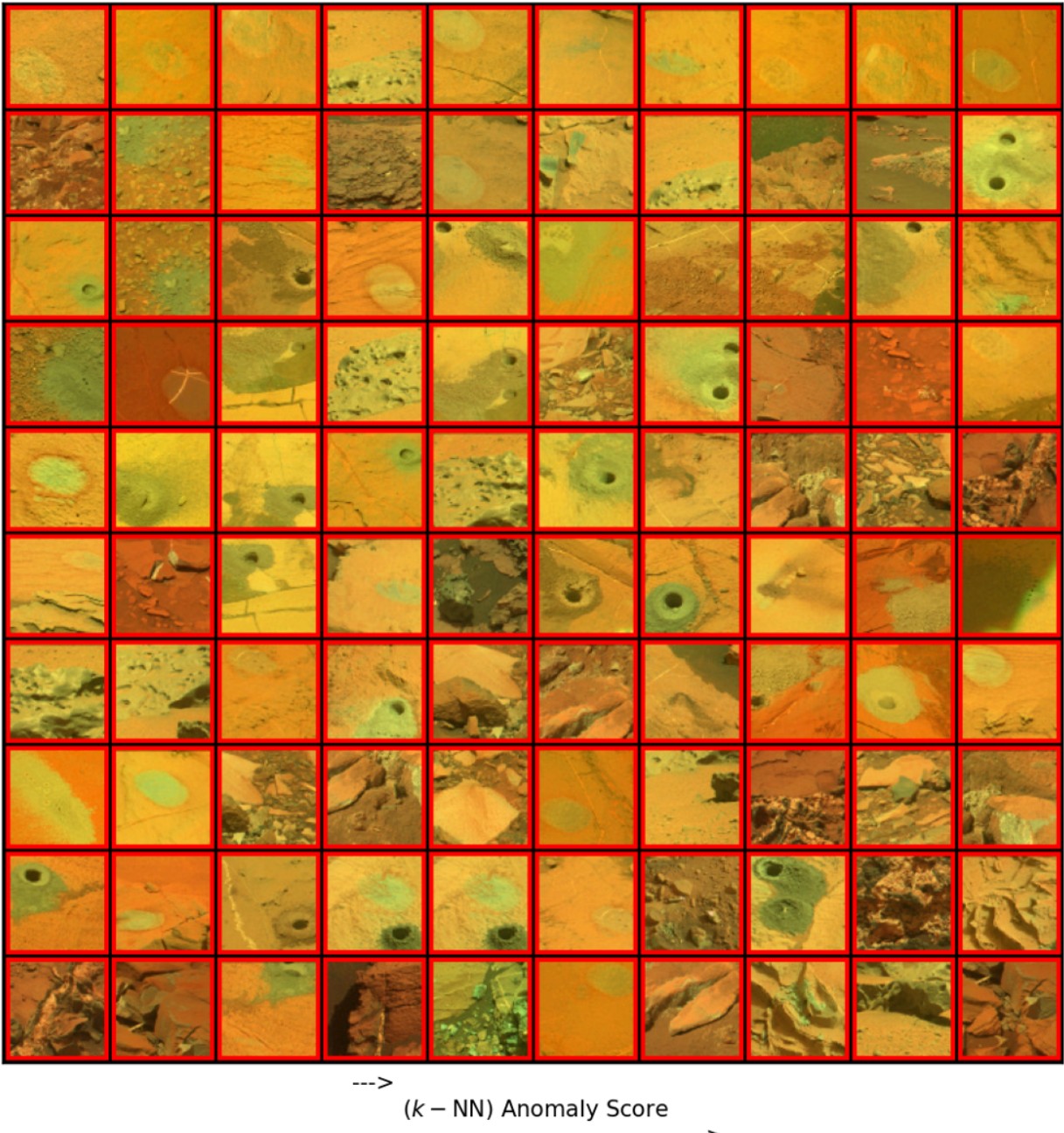

**Figure 21:** Results of the compressed VAE training, on the test set of the Mars Rover Mastcam dataset. Showing the images from the Novel test set only but with the index sorted by the anomaly score (from low to high). Only displaying one every 4 images, starting from index 0. Note how, in contrast to Figure 15, the samples with the lowest scores are not exclusively populated by plain textures like the 'drt' anomaly type, but also by more rocky textures, thing which makes the 'drt' anomaly type more difficult to conflate with similar plain textures in the normal class (and this is indeed reflected in Figure 18).

### 5.1.3 Standard VAE on the Galaxy Zoo dataset

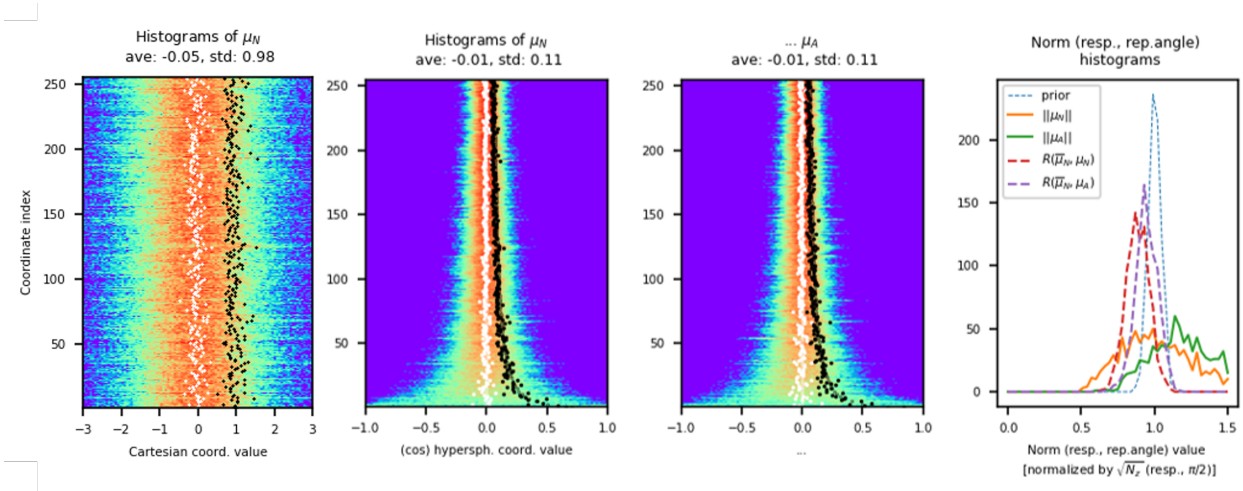

**Figure 22:** Results of the standard VAE training, on the test set of the Galaxy Zoo dataset.

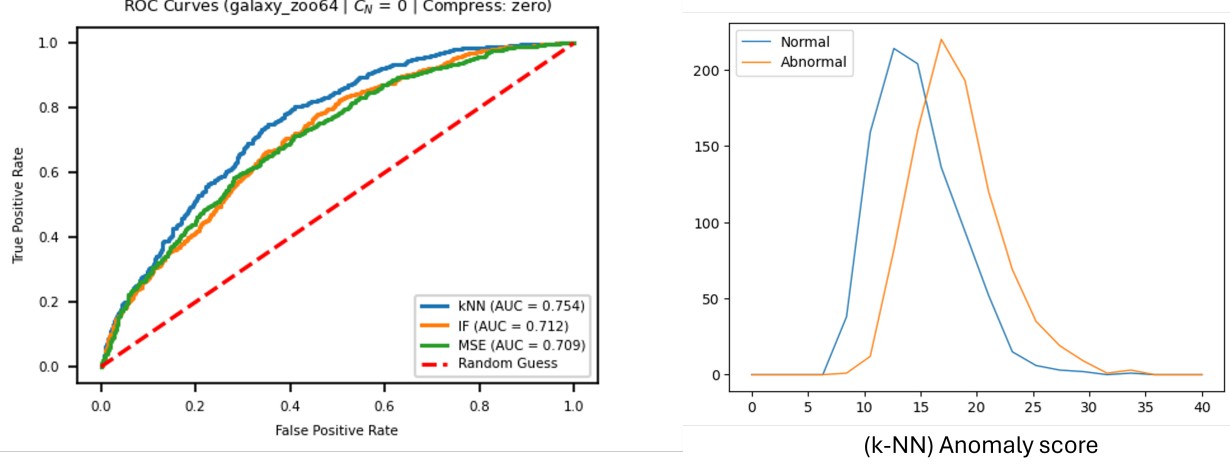

**Figure 23:** ROC curves and anomaly score histograms, results of the standard VAE training, on the test set of the Galaxy Zoo dataset.

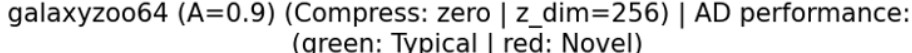

**Figure 24:** Results of the standard VAE training, on the test set of the Galaxy Zoo dataset. Showing the images from the test set but with the index sorted by the anomaly score (from low at the top to high at the bottom). Only displaying one every 20 images, starting from index 0. There are 1000 typical images and 1000 novel images in the testing set, for a total of 2000. A perfect detection would show the top half of the total panel as normal (green framing) and the anomalies (red framing) at the bottom half.

galaxyzoo64 (A=0.9) (Compress: zero | z_dim=256) | AD performance:
(showing only Typical)

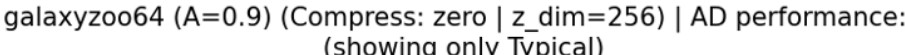

--->

$(k - NN)$ Anomaly Score

--->

**Figure 25:** Results of the standard VAE training, on the test set of the Galaxy Zoo dataset. Showing the images from the Typical test set only but with the index sorted by the anomaly score (from low to high). Only displaying one every 10 images, starting from index 0.

galaxyzoo64 (A=0.9) (Compress: zero | z_dim=256) | AD performance:
(showing only Novel)

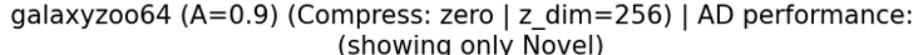

--->
$(k-\text{NN})$ Anomaly Score
--->

**Figure 26:** Results of the standard VAE training, on the test set of the Galaxy Zoo dataset. Showing the images from the Novel test set only but with the index sorted by the anomaly score (from low to high). Only displaying one every 10 images, starting from index 0.

### 5.1.4 Comp.VAE on the Galaxy Zoo dataset

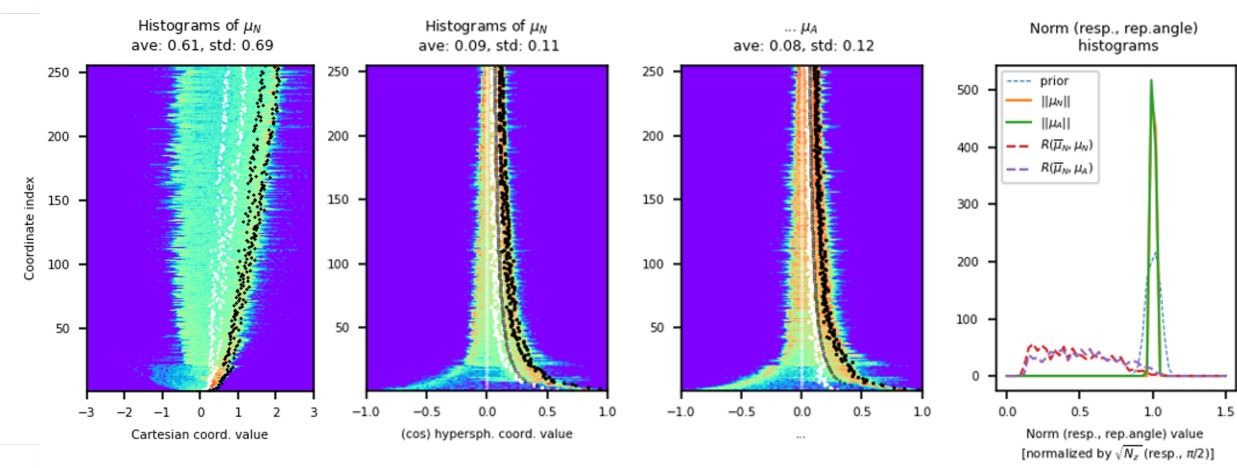

**Figure 27:** Results of the compressed VAE training, on the test set of the Galaxy Zoo dataset. Note how the means for the cosines of the hyperspherical angles are all shifted towards 1.0, but without collapsing the distribution. The compression direction in this case is given by the vector whose Cartesian coordinates are $(1, 1, ..., 1)$ (displayed as faint black dots in the cosine hyperspherical coordinates histograms for reference).

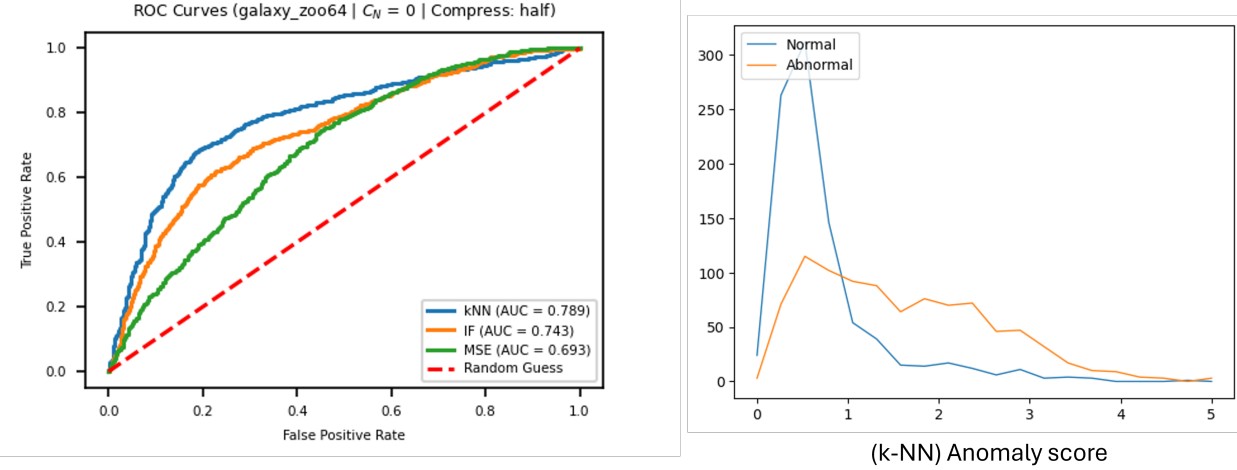

**Figure 28:** ROC curves and anomaly score histograms, results of the compressed VAE training, on the test set of the Galaxy Zoo dataset.

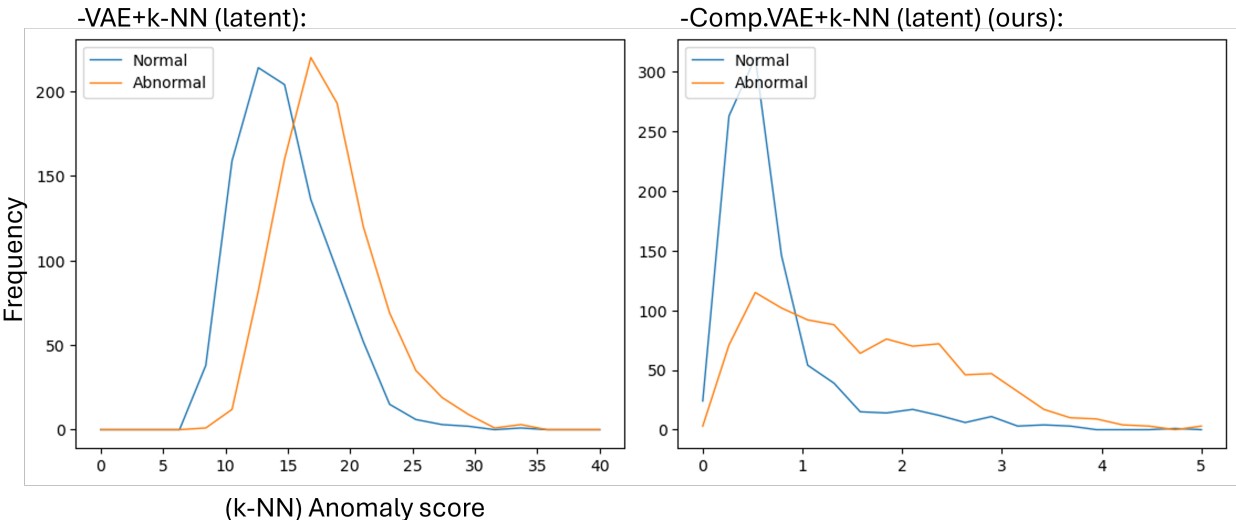

**Figure 29:** Note the impact of the volume compression of the normal class on the frequency of the $k-$NN anomaly score in the Galaxy Zoo dataset experiment: note the reduced value range in the right figure and the dense and compact population of the values close to 0 by only the normal class, while the anomalies are pushed to an elongated long tail (which can also be seen in Figure 1 from the paper). This results in better AUROC curve values (cf. Table 1, third column, from the paper). Note also the concentration of measure effects on the anomaly score for the standard AE (left figure): the normal class is concentrated around a mode which is far away from zero. There is a basic minimal distance, of around 5, under which no pair of normal samples can be found. This is because the vast volume of the equators, where these samples are located: to fall closer to each other is simply very unlikely, given that there is some much additional space to fall into.

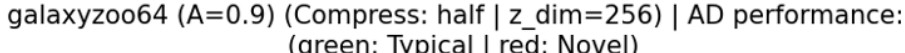

**Figure 30:** Results of the compressed VAE training, on the test set of the Galaxy Zoo dataset. Showing the images from the test set but with the index sorted by the anomaly score (from low at the top to high at the bottom). Only displaying one every 20 images, starting from index 0. There are 1000 typical images and 1000 novel images in the testing set, for a total of 2000. A perfect detection would show the top half of the total panel as normal (green framing) and the anomalies (red framing) at the bottom half.

galaxyzoo64 (A=0.9) (Compress: half | z_dim=256) | AD performance:
(showing only Typical)

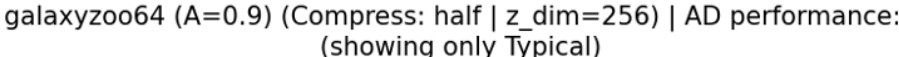

--->
$(k - NN)$ Anomaly Score
--->

**Figure 31:** Results of the compressed VAE training, on the test set of the Galaxy Zoo dataset. Showing the images from the Typical test set only but with the index sorted by the anomaly score (from low to high). Only displaying one every 10 images, starting from index 0.

galaxyzoo64 (A=0.9) (Compress: half | z_dim=256) | AD performance:
(showing only Novel)

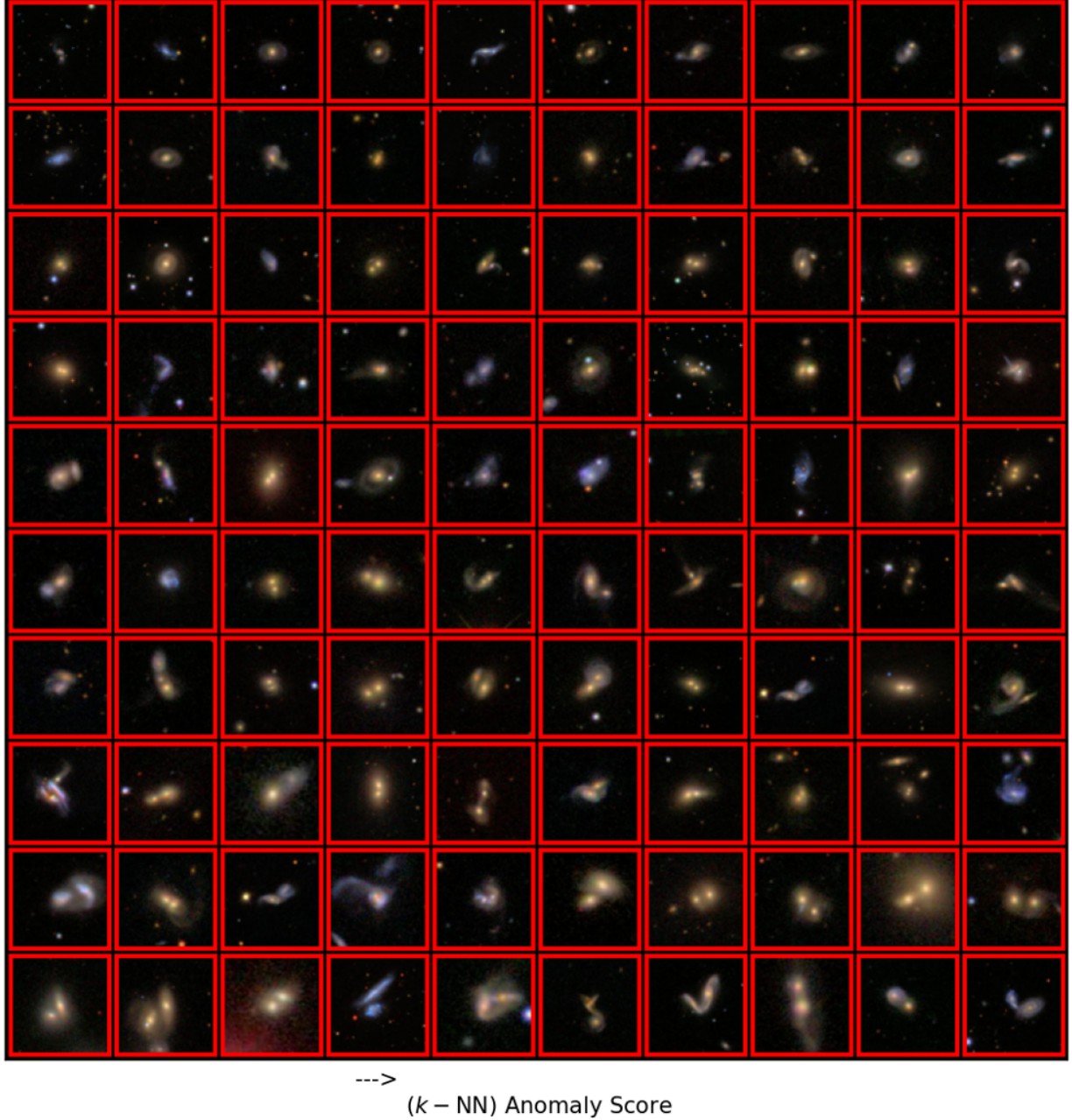

---> 
(*k* − NN) Anomaly Score
--->

**Figure 32:** Results of the compressed VAE training, on the test set of the Galaxy Zoo dataset. Showing the images from the Novel test set only but with the index sorted by the anomaly score (from low to high). Only displaying one every 10 images, starting from index 0.

### 5.1.5 Comp.VAE (vMF) on the Galaxy Zoo dataset

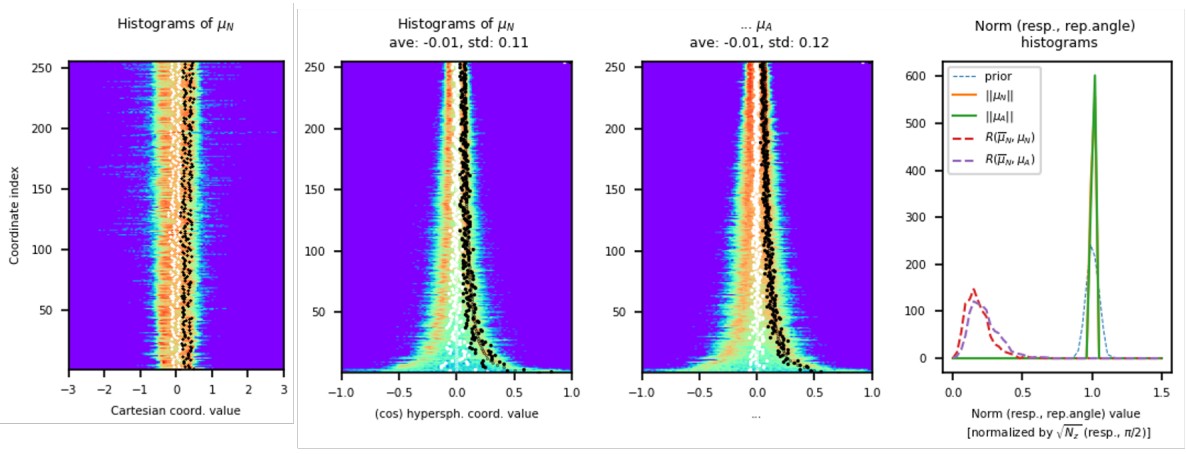

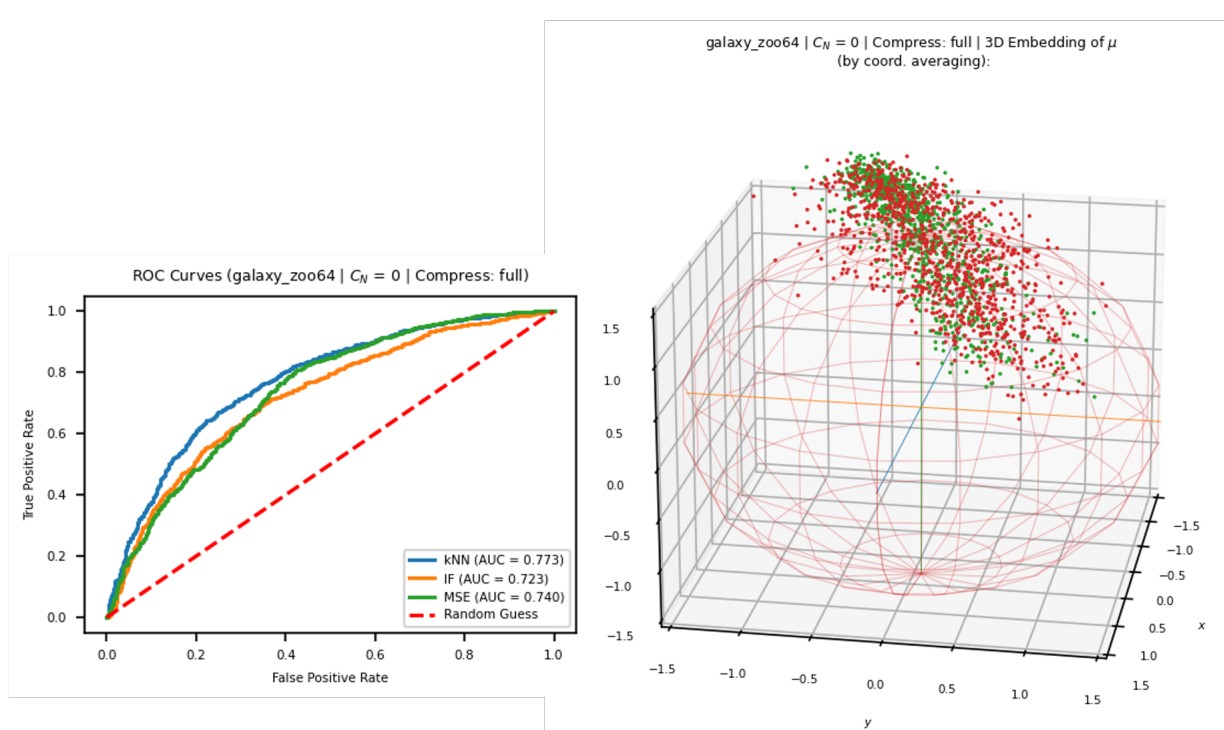

**Figure 33:** c.f. Figs.27, 28, and Fig.1 in the main paper.

## 5.2 Semi-supervised experiments

### 5.2.1 CIFAR-10 (ID) vs CIFAR-100

**CIFAR-10 ID Training: Comp.VAE-vMF model**

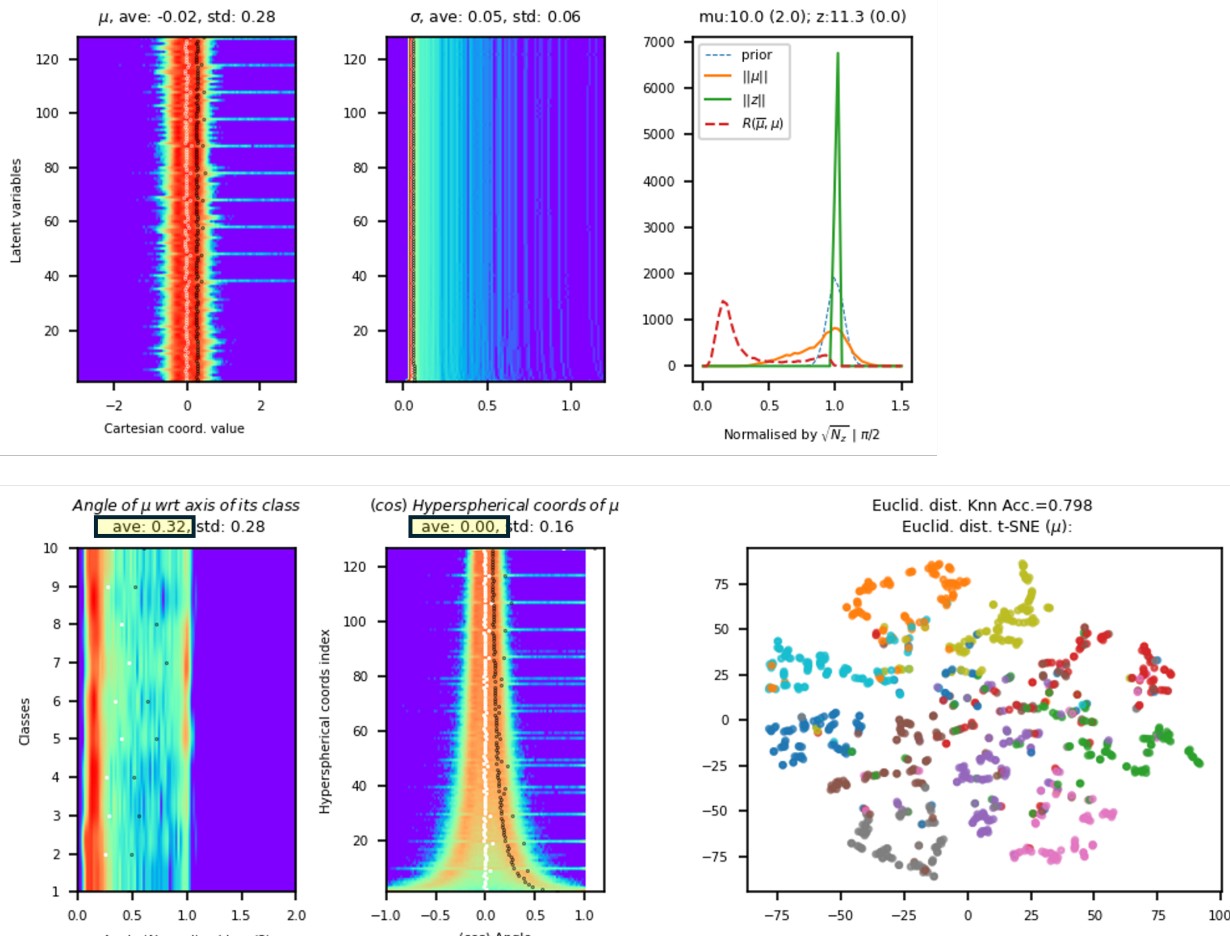

**Figure 34:** Upper panel, Cartesian coordinates histograms of $\mu$, $\sigma$, and the norm of $\mu$ and replica angle. Bottom panel, histograms for the first hyperspherical coordinate w.r.t. to the Cartesian axis corresponding to each sample's label, then the rest of the hyperspherical coordinates, and a t-SNE of the ID testing set latent embeddings. We highlight the average of the first angle (in the vMF approach, only this angle is compressed) and the average of the remaining hyperspherical coordinates (in our approach, all angles are compressed).

**CIFAR-10 ID Training: Comp.VAE-full compression model**

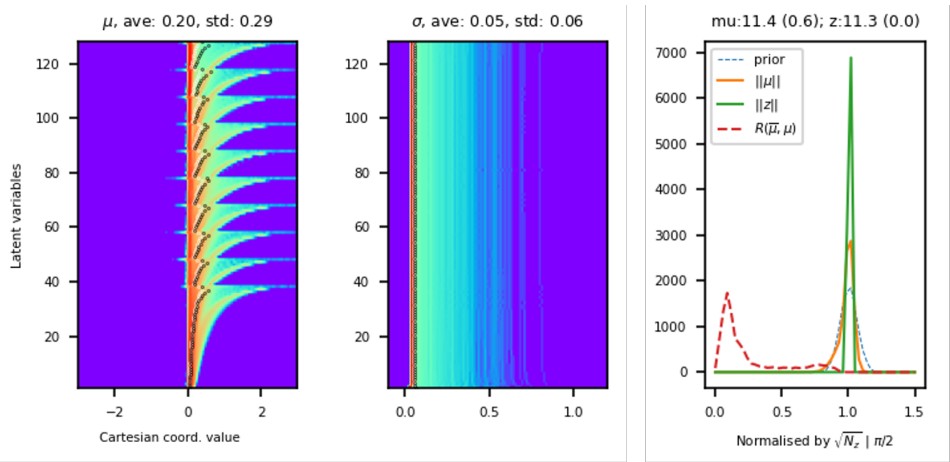

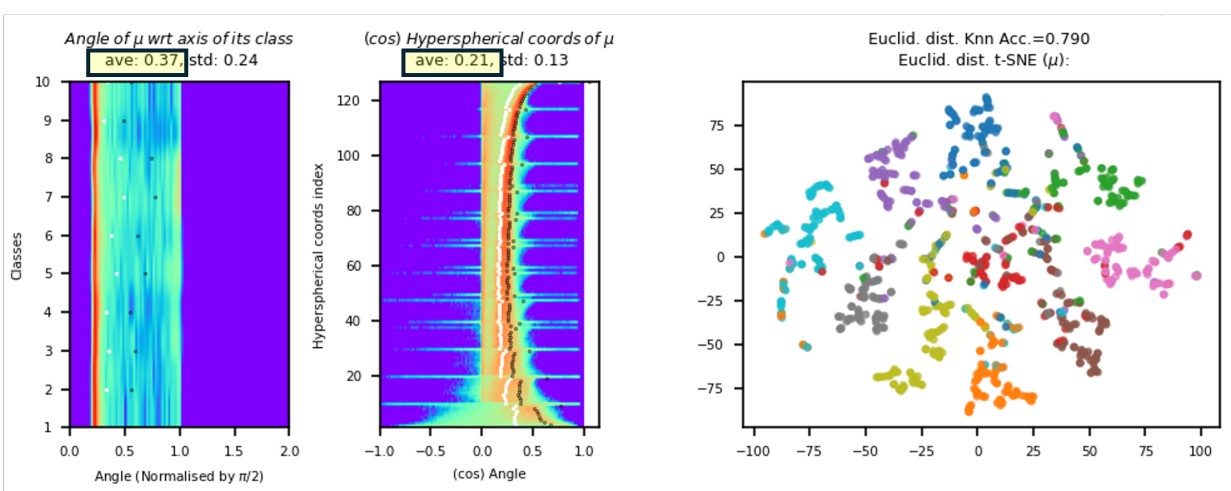

**Figure 35:** c.f. previous figure.

# CIFAR-100 near-OOD: comparison

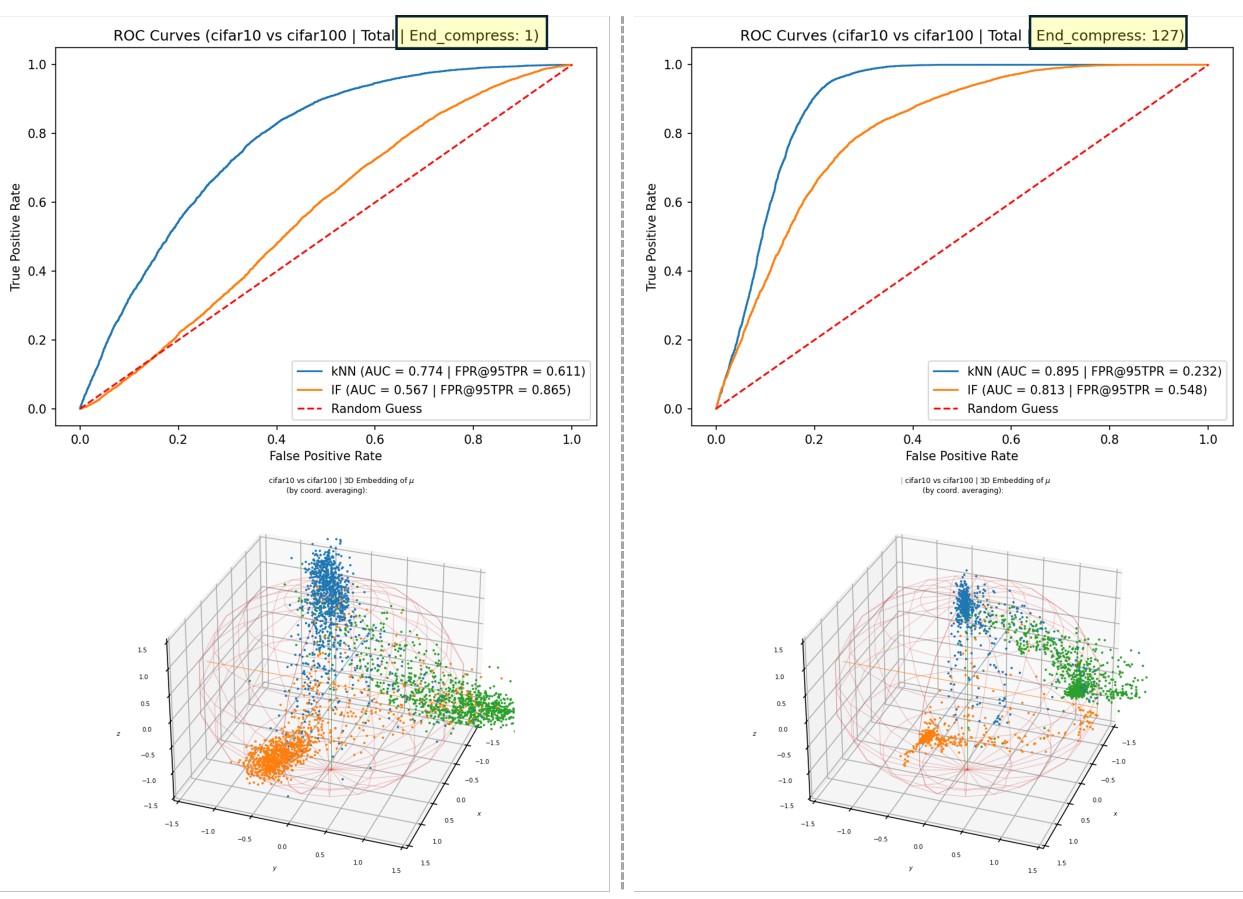

**Figure 36:** ROC curves and 3d visualizations. Left: Comp.VAE-vMF. Right: Comp.VAE full compression.

### 5.2.2 Imagenette (ID) vs close ImageNet classes

**Imagenette ID Training: Comp.VAE-vMF model**

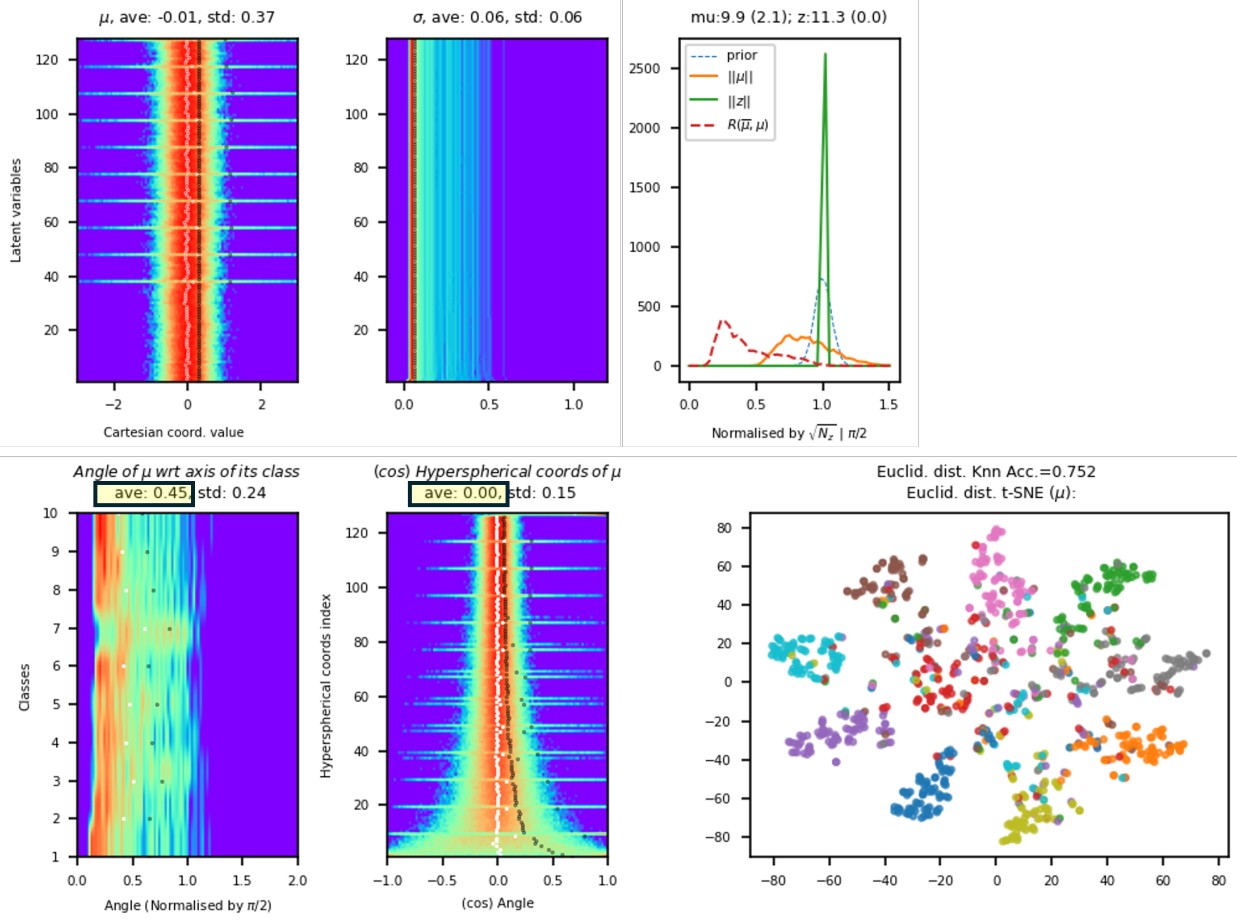

**Figure 37:** Upper panel, Cartesian coordinates histograms of $\mu$, $\sigma$, and the norm of $\mu$ and replica angle. Bottom panel, histograms for the first hyperspherical coordinate w.r.t. to the Cartesian axis corresponding to each sample's label, then the rest of the hyperspherical coordinates, and a t-SNE of the ID testing set latent embeddings. We highlight the average of the first angle (in the vMF approach, only this angle is compressed) and the average of the remaining hyperspherical coordinates (in our approach, all angles are compressed).

# Imagenette ID Training: Comp.VAE-full compression model

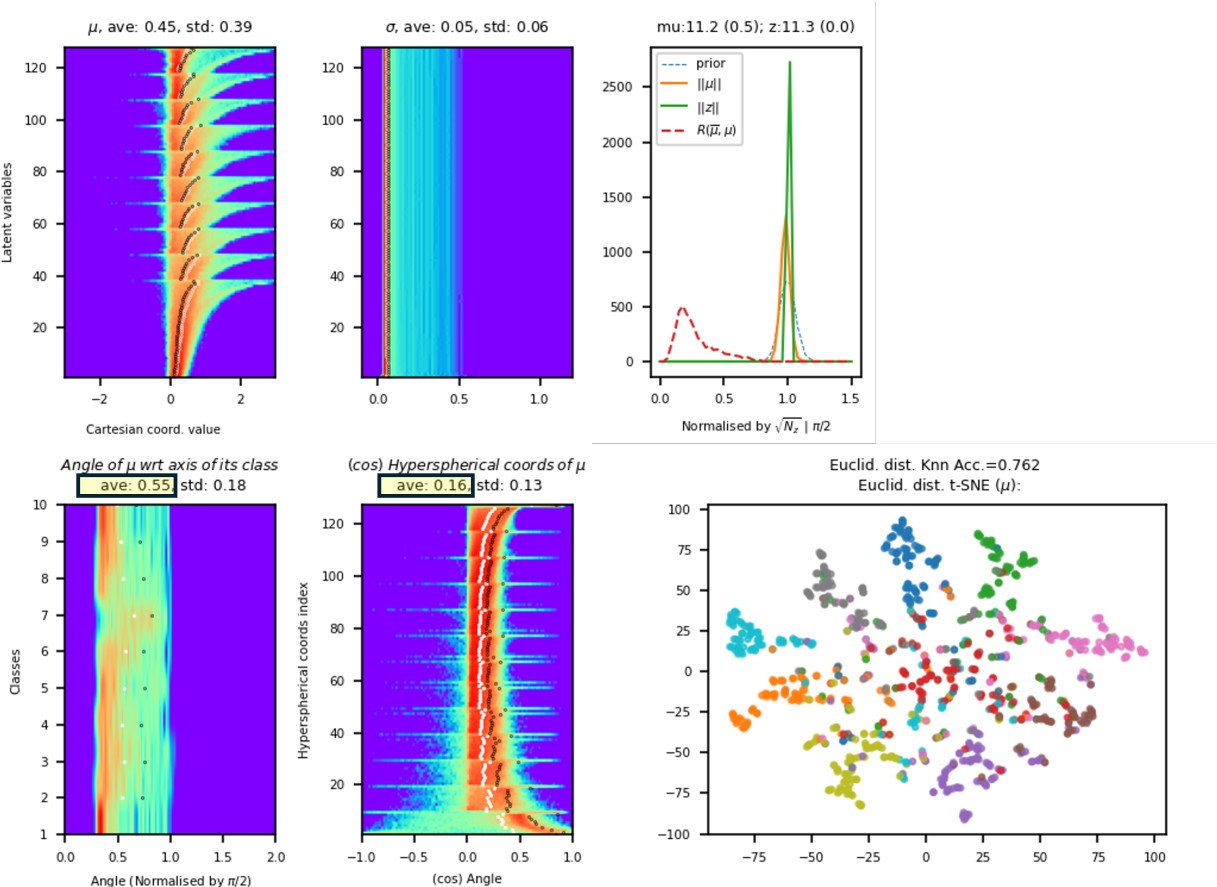

**Figure 38:** c.f. previous figure.

**close ImageNet near-OOD: comparison**

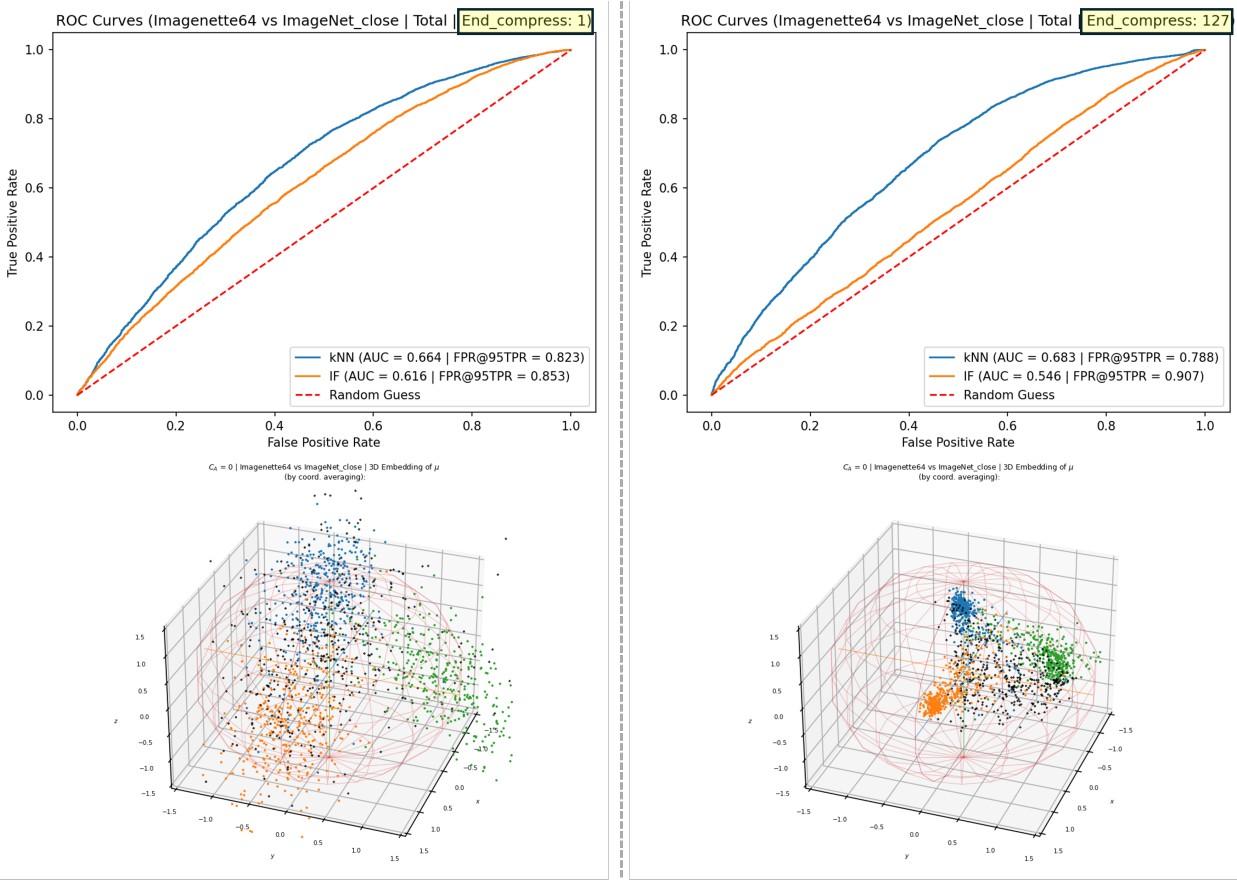

**Figure 39:** ROC curves and 3d visualizations. Left: Comp.VAE-vMF. Right: Comp.VAE full compression.

# 6  Results on CIFAR10 9-to-1 fully unsupervised AD

Another issue we encountered in our investigation on AD is the use of simulated anomalies to measure performance in the existing literature [8–10]. A typical experiment consists of using standard classification benchmark datasets (e.g., CIFAR10 or SVHN), and using one as the training normal dataset while another is the anomaly to be detected. Most methods perform very well (almost perfect detection) because, together with the semi-supervision, the abnormal samples are very different from the normal ones. An alternative is to use only one dataset, but label some classes as normal, while the remaining classes are the anomaly. This suffers from the opposite problem: in the fully unsupervised case there is no difference between normal and abnormal, and often many normal samples are more abnormal than abnormal classes. We provide evidence of these issues below, and advise against such experimental methods in the fully unsupervised case (the semi-supervised case performs well in the literature, cf. previous references).

When one class is used as normal (1-to-9), many of the datasets in the AD literature that we reviewed are too small (e.g., CIFAR10 has 5000 training samples per class) to properly train a model. Correcting the prevalence by using 9 classes as normal (9-to-1) results, in the fully unsupervised case, in that the abnormal data distribution overlaps with the normal, giving very low performance (AUROC curve value $\sim 50\%$ in average across all the classes at best). In fact, when looking at the performance per class, the AUC is often below $50\%$, revealing an inversion of the classification. In other words, the abnormal class becomes closer to the normal one than most of the normal samples, probably due to the complex topology and geometry of the class manifold, likely related to the randomly changing backgrounds in most of the images, which is not the case in true anomaly detection, when the normal class can be somewhat complex, but should be predictable. Once this is in place, one could study the effect of randomly changing backgrounds, but that is an addition, a posteriori task to basic AD.

We observed this phenomenon whether the anomaly scores are computed from the original data or their latent (via an AE or VAE), in all the methods that we tested.

Here we include the results of our 9-to-1 AD experiments on pixel-space for CIFAR10 (that is, 9 classes are selected as normal while the remaining one is taken as the source of anomalies during testing). Note that what we performed are the fully unsupervised version of these experiments, that is, the subclass information for the normal distribution (composed of 9 subclasses, as mentioned) is not used at any point, contrary to many experiments in the literature which do take this information into account, thing which is used to disentangle the normal subclasses with a classifier (e.g., a ResNet) and then the obtained 'good' embeddings are used to perform the AD, this greatly simplifies the problem.

As can be seen in the figures below, the results can be quite inconsistent when varying the anomalous class, since, for some of them, one can observe pathological behaviors, like the inversion of the ROC curves (which means that the testing anomalous class is closer to the training normal than the testing normal class itself, as the histograms for the anomaly scores confirm).

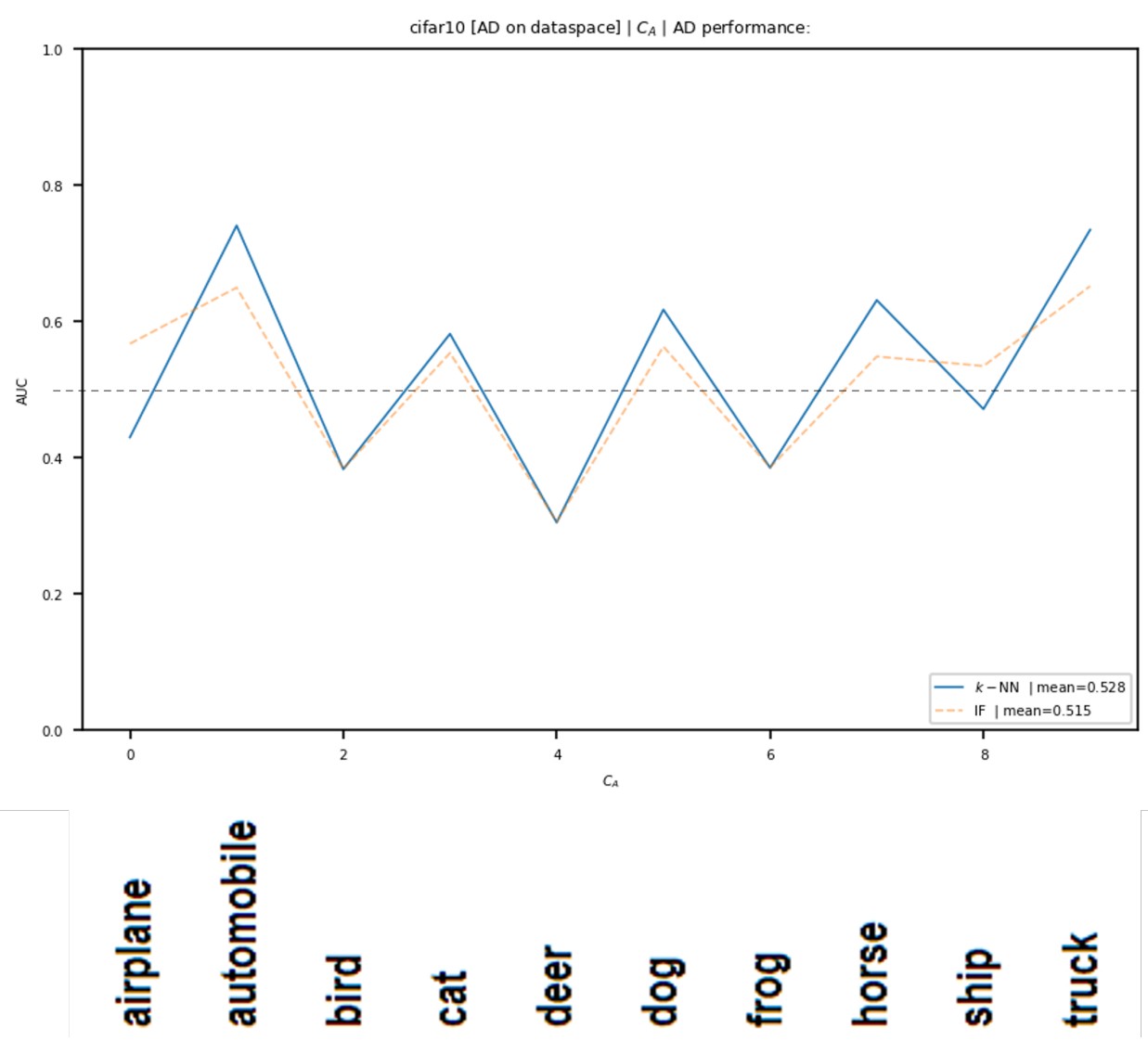

**Figure 40:** AUROC curve values for 9-to-1 AD experiments on pixel-space for CIFAR10. $C_A$ in the horizontal axis indicates the class being taken as anomalous. The grey horizontal dashed line corresponds to the AUC value of 0.50.

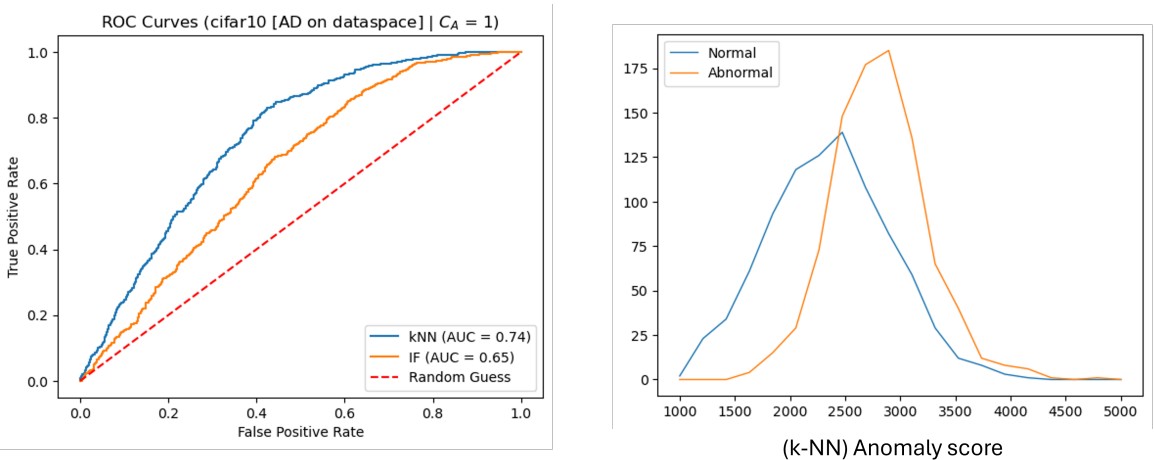

**Figure 41:** ROC curves and anomaly score histograms, results for the $C_A = automobile$ experiment.

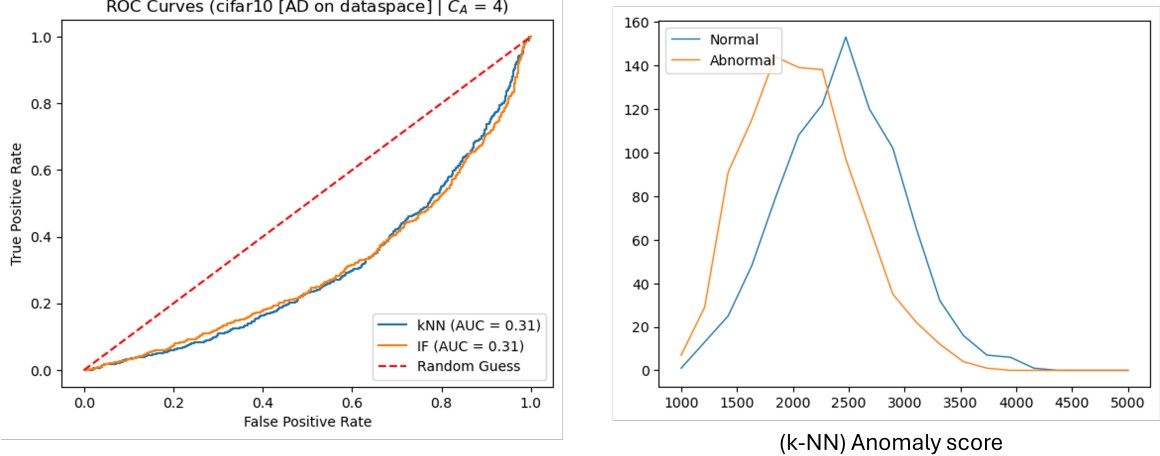

**Figure 42:** ROC curves and anomaly score histograms, results for the $C_A = deer$ experiment.

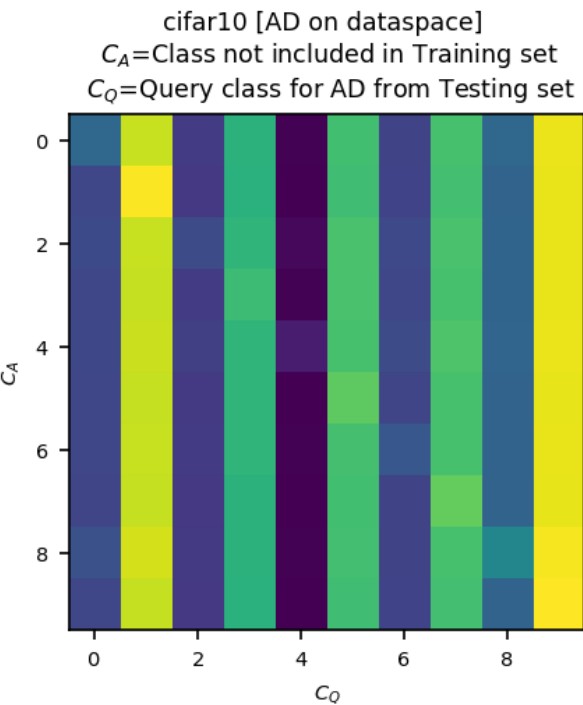

**Figure 43:** Matrix with all the k-NN anomaly score values (blue means low, yellow is high). The horizontal direction is one single 9-to-1 experiment. In a better behaved case, one would expect the diagonal to be yellow while the rest of the matrix to be blue.

Even if one employs a new model or method capable of re-inverting scores from baseline methods to recover standard AUC behavior in those test anomaly classes, this does not resolve the underlying issue: the dataset itself becomes unreliable due to this inversion effect. This unreliability undermines confidence in the evaluation process, especially when encountering *new or previously unseen anomaly types/classes.* In such cases, one cannot determine whether a low anomaly score corresponds to an actual anomaly or a misclassified normal sample, since *the inversion depends on the anomaly type/class being considered* (as Fig.40 shows), and the resolution of the inversion problem is only confirmed for the known types. Of course, this could also happen for a dataset which did not have the inversion problem in the first place for the known classes, since we cannot known what the behavior will be for new anomaly types. But we simply cannot do anything to remedy that, while we can do something to remedy the former inversion case described here.

If a newly constructed dataset from a real world application yields inverted AUC values for standard baseline methods, this should be treated as a red flag. It suggests that the dataset may not be structurally sound for reliable anomaly detection, and should be reconsidered or refined to ensure that the scoring behavior is consistent and interpretable across all test sample types. One needs solid and stable baselines and datasets to properly perform fully unsupervised AD.

The bottom line is that the main problem with the use datasets which are known to give inverted AUC metrics (i.e., where lower anomaly scores correspond to more anomalous samples) for basic baselines is the ambiguity it introduces in interpreting individual results in new methods. For a new test sample whose anomaly score is lower than those of most normal samples, it is unclear whether the sample is a true anomaly (under an inverted scoring convention), or simply a normal instance that lies in the left tail of the normal distribution.

# 7 Model Details and Implementation

## 7.1 Model

See Table.2. Each residual block contains two Conv+BN+LeakyReLU layers, with optional expansion layers.

## 7.2 Choosing the gain for each loss term

The constants $\alpha_{i,j}$, $\beta_{i,j}$ multiplying the elements of the hyperspherical loss are proportional to $1/\sqrt{k+1}$, where $k$ is the coordinate index. This was necessary because, unlike the Cartesian coordinates, the hyperspherical coordinates are asymmetric and vary with $k$. This can be seen in the transformation formulas, where a product of an increasing amount of sine functions is necessary as the coordinate index increases. We chose $1/\sqrt{k+1}$, guided by the fact that the vector whose Cartesian coordinates are $(1, 1, ..., 1)$ has a cosine of its spherical angles equal to $1/\sqrt{k+1}$ as the coordinate index k varies, and because it gave the best results experimentally.

In this way, we were able to avoid lengthy calculations to obtain the mathematically exact formulas for both these constants and the KLD in hyperspherical coordinates, which we do not believe to be important for the goals of this work.

Finally, there are single scalar hyperparameters/gains multiplying each loss part after summing up the corresponding index $k$ in each of them, and a global single scalar hyperparameter/gain $\beta$ multiplying the total KLD-like loss in hyperspherical coordinates. The optimal value for these hyperparameters was found via a simple grid search and are provided in the full code release.

**Table 2:** Architecture of the proposed Variational Autoencoder (VAE).

| Component | Layer Details |
| --- | --- |
| **Encoder** | |
| Input | $3 \times 64 \times 64$ RGB image |
| Conv Block | Conv(3,16,5,pad=2) $\rightarrow$ BN $\rightarrow$ LeakyReLU $\rightarrow$ AvgPool(2) |
| Residual Block @32 | ResidualBlock(16$\rightarrow$32) |
| Downsample | AvgPool(2) |
| Residual Block @16 | ResidualBlock(32$\rightarrow$64) |
| Downsample | AvgPool(2) |
| Residual Block @8 | ResidualBlock(64$\rightarrow$64) |
| Flatten + FC | Linear(4096$\rightarrow$512) |
| **Latent Space** | $z \in \mathbb{R}^{256}$ (via reparameterization) |
| **Decoder** | |
| FC + Activation | Linear(256$\rightarrow$4096) $\rightarrow$ BN $\rightarrow$ LeakyReLU |
| Residual Block @4 | ResidualBlock(64$\rightarrow$64) |
| Upsample | $\times 2$ |
| Residual Block @8 | ResidualBlock(64$\rightarrow$32) |
| Upsample | $\times 2$ |
| Residual Block @16 | ResidualBlock(32$\rightarrow$16) |
| Upsample | $\times 2$ |
| Residual Block @32 | ResidualBlock(16$\rightarrow$16) |
| Output | Conv(16,3,5,pad=2) $\rightarrow$ Sigmoid |
| **Total Parameters** | 3,422,979 |

## 7.3   Differences in training speed

We provide here data regarding the differences in the training speeds between the standard VAE and our compression VAE via hyperspherical coordinates. The origin of this difference mainly lies in the extra calculations needed for the coordinate transformations in 1.1, which are implemented via the script in 1.2.

The measurements were done during typical trainings in a NVIDIA H100 GPU. In Fig.44 we show the results for the case of trainings with CIFAR10, with a batch size of 200 samples, and the changes in training speed (measured as how many batches per second are being processed) in terms of the dimension $n$ of the latent space. After $n = 200$, until $n = 800$, the decay is almost linear in $n$, with a decay rate in the speed of 20 batch/s every 200 latent dimensions.

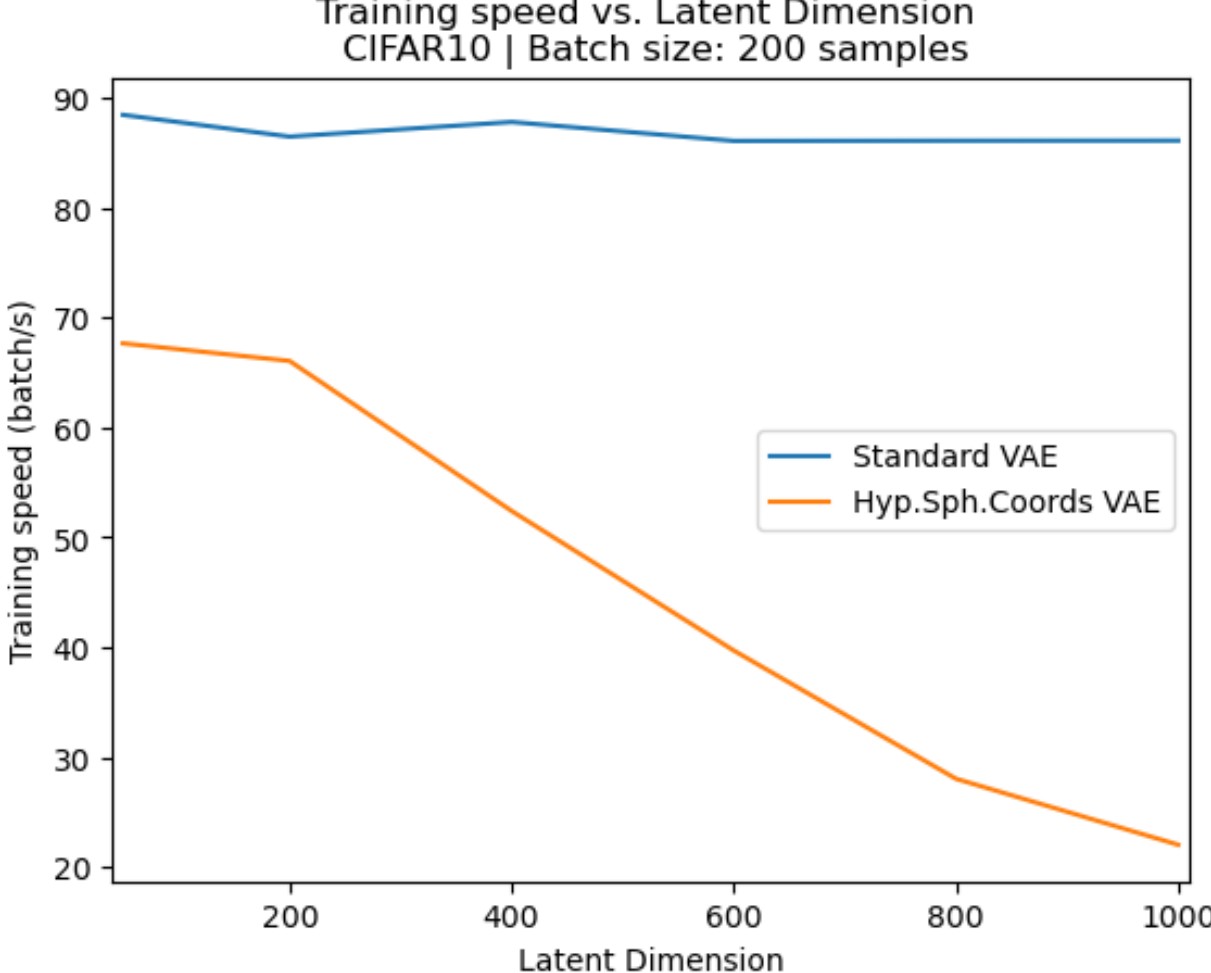

**Figure 44:** Differences in training speed.