# OpenReview forum: "VAE with Hyperspherical Coordinates: Improving Anomaly Detection from Hypervolume-Compressed Latent Space"
_ICLR.cc/2026/Conference — Submitted to ICLR 2026_

### Official Review · Reviewer_whHn · 2025-10-27

**Soundness:** 3
**Presentation:** 2
**Contribution:** 2
**Rating:** 2
**Confidence:** 4

**Summary:**

This paper proposes a method to compress the latent space of VAE for learning more distingushiable representations for AD tasks. Specially, the authors first convert the encoder output from Cartesian coordinates into hyperspherical coordinates,
then the prior is Specially designed so that samples can be moved to a zone with reduced volume. Experiments conducted on several image datasets illustrate the effectiveness of the proposed method.

**Strengths:**

- The experiments are conducted under both unsupervised and semi-supervised settings.
- Overall the paper is well-structured and easy to follow, except several parts of 2.
- The proposed method is well-motivated.

**Weaknesses:**

- Questionable novelty. The core technique appears highly similar to the method in [1], and the paper does not clearly delineate what is new.
- Incomplete related work. Prior studies on concentration-of-measure phenomena and compression (e.g., [2]) are not discussed, nor is [2] used as a baseline.
- Weak baselines. In the fully unsupervised setting, only several AE/VAE-based baselines are considered. Many recent unsupervised AD methods are missing (e.g., [3]–[5]).
- Missing large-scale benchmark. Evaluation on a widely used benchmark such as ADBench would better substantiate the method’s effectiveness and generality.
- Conceptual conflation. The paper treats OOD detection as semi-supervised AD. These are distinct problem formulations; see [6] for definitions of weakly/semi-supervised AD. Meanwhile, the proposed method is not effective in the OOD case.
- The usage of dimensionality reduction methods plus kNN is not novel and result in high time complexity of $\mathcal{O}(n^2)$ for inference, suppose $n$ is the number of samples in training dataset.

[1] Ascarate, Alejandro, et al. "Improving the Generation of VAEs with High Dimensional Latent Spaces by the use of Hyperspherical Coordinates." arXiv preprint arXiv:2507.15900 (2025).

[2] Zhang, Yunhe, et al. "Deep orthogonal hypersphere compression for anomaly detection." arXiv preprint arXiv:2302.06430 (2023).

[3] Livernoche, Victor, et al. "On diffusion modeling for anomaly detection." arXiv preprint arXiv:2305.18593 (2023).

[4] Yin, Jiaxin, et al. "MCM: Masked cell modeling for anomaly detection in tabular data." The Twelfth International Conference on Learning Representations. 2024.

[5] Thimonier, Hugo, et al. "Beyond individual input for deep anomaly detection on tabular data." arXiv preprint arXiv:2305.15121 (2023).

[6] Durani, Walid, et al. "Weakly Supervised Anomaly Detection via Dual-Tailed Kernel." Forty-second International Conference on Machine Learning.

**Questions:**

- See weaknesses.
- The derivation in this paper is not sufficiently clear. For example, how is equation (2) obtained from equation (1)? In equation (2), how are the coefficients α and 𝛽 introduced and defined?

---

> ### Author Response · Authors · 2025-11-20
> **"Overall the paper is well-structured and easy to follow"!**
>
> Weaknesses:
>
> 1. What is similar is the technique to compress volume, but that reference only showed the benefit for the generative task/ability of the VAE. We used that method but adapted it to anomaly detection, which is a completely different task. We do not know any work investigating anomaly detection within that framework.
>
> 2. Thank you for the reference which is relevant but not directly comparable. The suggested reference indeed studies one concentration of measure issue, namely the convergence of the Gaussian in very high dimensions towards a uniform distribution on the hypersphere. However, it proposes an anomaly detection method based on radial distance within the hypersphere. In contrast we consider anomaly using a distance that encompasses radial and angular distances: we detect how far anomaly are from a normal island, compared to the suggested reference measuring anomaly within the thickness of the hypersphere. We also note that a VAE operating in high dimensions has a very thin hyperspherical skin and the radial distance is very compressed. Additional experiments (not included for lack of space) show that radial variation affects mostly image contrast, not semantic information.  Our paper deals with the equatorial presentation of the hypervolume (i.e., not just on a sphere, but also only on its equators, for any randomly chosen north pole). This is key because it means that the equators must also be very sparse due to their high hypervolume, while this is not the case for the hemispheres. What we propose, and show experimentally, is that by moving samples away from the sparse region to the better-behaved hemispheres, machine learning tasks like anomaly detection get a considerable improvement.
>
>
> 3. The baselines considered in the paper were deliberately selected to highlight the impact of the equator issue on anomaly detection. Others have shown that the baseline we selected were state of the art for our applications (see references for the dataset). We therefore limited our comparison to those avoiding unnecessary use of resources. If requested, we could copy more baselines in our results from the relevant references.
>
>
> 4. For the OOD case, the experimental framework and datasets we used are standard and considered sufficient in many recent papers published on top tier venues, we are simply following that already established set up (see the references in our paper, many from previous ICLR). For the fully unsupervised case, we chose two complex, real world imaging datasets which also had enough training data. Unfortunately, this is not the case for many datasets often used in the literature, for example MVTec a popular imaging dataset has less than 5000 samples. ADBench is a terrific resource that we are actively using to explore our method for 1D anomaly detection. For images, we selected two very different types of dataset with real anomaly. Our results show a robust/systematic effect on improving the FPR95 (and AUC) across these datasets and for both fully unsupervised and OOD detection.
>
>
>
> 5. Yes, this is correct, we thank the reviewer for pointing this out. Our rationale for the choice of terminology was to make clear the distinction between these two different types of anomaly detection. But we agree that it may lead to confusion and so we will revert to the standard terminology in the paper. Our method does reach state-of-the-art at the metric FPR95 (false positive rate at 95% true positive rate), which is key for anomaly detection, for far-OOD in relation to comparable methods. For near-OOD our method shows excellent results, particularly in Cifar10 vs Cifar100.
>
>
>
> 6. The use of kNN on features for anomaly detection is certainly not new. In fact, we explicitly mention references with proposals in that line, as well as including this as baseline. The method we present is about measuring the impact of the equator issue and how the use of compressed representations can help to avoid it and improve anomaly detection. We chose kNN to operate on these representations since it is the simplest choice, but one could also use other methods (indeed, we also included comparisons with another non-parametric method in our paper, namely, Isolation Forest on the representations). The compression method is agnostic to the detection method.
>
>
>
> Question:
>
> The second equation is not derived from the first one. Instead, it is postulated by formal analogy and based on the effect we want it to achieve during training. Alpha and beta are free parameters or gains/weights for each corresponding part of the total loss, like the beta in a standard beta-VAE. The value of these parameters was determined by doing a grid search among many and only retaining for our model the best performing one. We will make sure that this is clearer in the revision.

---

> > ### Comment · Reviewer_whHn · 2025-11-26
> >
> > I thank the authors for their brief explanations. However, since most of my concerns remain unresolved, I maintain my current score.
> >
> > ### • Novelty
> > Directly applying techniques from another area without any adaptation for anomaly detection raises concerns regarding the motivation and contribution of the work.
> >
> > ### • Baselines
> > I checked the dataset references and noticed that the citation is more than five years old. It is unclear how such a reference could cover recent state-of-the-art methods. The authors are encouraged to provide the exact paper reference for clarity.
> > More importantly, I reiterate that additional recent baselines should be included for both UAD and OOD detection to convincingly demonstrate the effectiveness of the proposed method.
> >
> > ### • Dataset Selection
> > The two datasets used for UAD are not commonly adopted in recent literature, and the datasets selected for UAD and OOD detection differ completely. The authors should consider adding experiments on widely used datasets such as CIFAR-10 under the standard UAD setting (i.e., one-class classification).
> >
> > ### • Detection Method
> > Although the authors claim that *“the compression method is agnostic to the detection method”*, the results show a significant performance drop when using Isolation Forest. This observation appears inconsistent with the claim and warrants further clarification.
> >
> > ### • Reproducibility
> > No code is provided for reproducing the experimental results, which limits the reproducibility of the work.
> >
> > ### • Hyperparameter Tuning
> > In the response, the authors mention that the values of $\alpha$ and $\beta$ are selected via grid search by choosing the best-performing configuration. It is unclear whether a similar tuning strategy was applied to all baselines. If not, such selective tuning may artificially inflate the performance gains of the proposed approach.

---

### Official Review · Reviewer_gFF6 · 2025-10-29

**Soundness:** 2
**Presentation:** 2
**Contribution:** 3
**Rating:** 4
**Confidence:** 2

**Summary:**

The paper proposes a modification of the standard Variational Autoencoder (VAE) by introducing hyperspherical coordinates in the latent space and having adapted priors to these coordinates instad of Gaussian priors in euklidean space. The authors argue that this change addresses issues related to high-dimensional Gaussian distributions, such as concentration of measure and the tendency of latent vectors to cluster in equatorial regions of the hypersphere. They apply their method to anomaly detection in both unsupervised and semi-supervised settings and evaluate it on multiple datasets.

**Strengths:**

1. The paper addresses an interesting conceptual idea: leveraging hyperspherical geometry to restructure the latent space of VAEs.
2. The proposed approach is technically novel in how it reformulates the latent distribution.
3. The experimental results look strong on first sight.
4. The appendix is really strong and includes a lot of intuitoin and background information (which would have helped in understanding the paper).

**Weaknesses:**

1. Lack of statistical rigor in evaluation:
The results section does not include any indication of how many runs were performed per method, nor are any measures of statistical variation (e.g., standard deviation, confidence intervals) reported. This is a major issue.
2. Unmotivated choice of high-dimensional latent space:
The paper uses high-dimensional latent vectors (e.g., 100–256 dimensions) but provides no strong empirical or theoretical justification for this choice. It is unclear whether lower-dimensional latent spaces perform worse or whether the method’s advantages only emerge in high dimensions.
3. Limited methodological accessibility:
While the paper touches on complex geometric ideas such as volume concentration on hyperspheres and almost-orthogonality, these are presented without sufficient intuition or formal support. Notation is moved to the appendix. Readers unfamiliar with high-dimensional geometry will find it difficult to follow key parts of the method. More explanatory figures, examples, or visualizations would have helped greatly. I also do not understand what $a_{i,j}$ and $b_{i,j}$ are from the short description (even though I can make a probably correct educated guess).
4. Unclear problem significance:
The paper assumes that measure concentration and “equator clustering” are problematic in standard VAEs for anomaly detection, but it does not provide convincing evidence that this is a real or significant problem in practice. No baseline analysis is presented to demonstrate that the equator effect harms anomaly detection performance. As such, the motivation for the proposed fix remains speculative.
5. Notational problems (sometimes due to notation only being explained in the appendix): E.g. in (2) on the left hand side, you have $\varphi_k$ for a fixed $k$ and on the right hand side $k$ is a summation index. Or why does setting a mean in (6) enforce a radius without fixing a standard deviation to zero? Also, the citations are formated wrongly in the paper text.

**Questions:**

Does your prior have any classical description? I can (I think) understand the formulas and I can mostly follow your argument for the prior, but I am lacking a clearer understanding in terms of math.

---

> ### Author Response · Authors · 2025-11-20
> **"The paper addresses an interesting conceptual idea"!**
>
> Weaknesses:
>
> 1. Thanks for pointing this out, we will revise our manuscript to include this information.
>
> 2. Rich and complex images like the ones used in this paper need relatively high dimensional latent spaces in order to fully capture all of the relevant features. In any case, when running our experiments, we performed a grid search for the values of the dimension and the parameter beta. The ones described in the paper are the values with the best performances. Lower dimensions consistently degraded the anomaly detection performance. The 256 dimensions of latent is typical for this type of data.
>
> 3. These are complex topics and due to space we cannot discuss it at length in the main paper. We provided in the supplementary material exactly the kind of requested explanatory figures and visualizations in order to quickly gain intuition about these phenomena. The values a,b are the target/prior values. We want the latent means and stds to converge to those priors during training; one simply specifies them, this can also be done in the standard VAE (e.g., centring the prior somewhere else rather than on the origin).
>
> 4. We included in all our tables the performance of the standard VAE as a baseline, which optimize the latent to be normally distributed. In high dimensions, a multivariate Gaussian distribution (latent prior) is similar to a uniform distribution on the hypersphere, and thus the probability mass is exponentially concentrated on equatorial bands. This can be appreciated clearly in the histograms for the angles and hyperspherical coordinates provided in the supplementary material.
>
> 5. Yes, we do set the std of the radius to zero in most of our experiments. This can be seen in the sharp radius histograms in the supplementary, but it is true that it was not mentioned in the main paper. We will correct this, thank you for pointing this out. Regarding the sub index k, the one in the left is just to indicate that the loss is dependent on a vectorial quantity (here all the hyperspherical coordinates angles), it’s an abuse of notation yet fairly standard; on the right it indeed is a summation, we have similar terms for each k which are then summed over k.
>
>
> Questions:
>
> For 3D Euclidean space, one can imagine a standard spherical coordinate system where the points are forced to move and accumulate near the north pole by means of forcing their spherical coordinates angles towards zero. But this picture can be misleading in high dimensions since the north pole in that case is almost impossible to reach (it has exponentially small probability weight), and the samples tend to form rings around the south-north axis. This is actually good, since the north pole is singular for spherical coordinates. As for the math, only the coordinates transformation formulas detailed in the supplementary is all of what is really needed. It also explains why this is beneficial to hypervolume reduction by analysing the structure of the hypervolume element when expressed in this coordinate system. The latter is the mathematical origin/explanation of why/how our method works in its intended purpose. Another way of seeing it in 3D is the fact that the coordinate lines of the latitude angle (pick any longitude, move only in latitude) converge to the north pole, hence a small square of fixed angular_coordinates-size will have less metric area by just moving it closer to the north pole without changing anything else. This is because the north pole is what in geometry is called a `conjugate point’ of the south pole, due to the curvature of the sphere.
>
>
> In a physics analogy, if the latent space is considered to be the configuration space of a very large number of small magnets, the Gaussian case corresponds to a high temperature system with no global magnetization (called a "disordered" state/high entropy), while our compressed VAE latent corresponds to a low temperature "ordered"/low entropy state with a characteristic nonzero global magnetization. Thus, in effect, our prior is forcing the system to make the phase transition from disordered to ordered.

---

> > ### Comment · Reviewer_gFF6 · 2025-11-22
> >
> > I have read the other reviews and the rebuttals. In particular, reviewer 3 makes good points. I have not changed my opinion about this paper.

---

> ### Author Response · Authors · 2025-11-25
>
> Can we ask you to elaborate? Since you said you read all our answers to all reviewers and still have the same opinion, then which aspect of our answers you didn't find satisfactory? Thanks.

---

### Official Review · Reviewer_8nMQ · 2025-11-01

**Soundness:** 2
**Presentation:** 2
**Contribution:** 2
**Rating:** 4
**Confidence:** 3

**Summary:**

This paper proposes a hyperspherical-coordinate reformulation of the VAE to mitigate high-dimensional sparsity in latent space. By compressing latent vectors toward compact regions on the hypersphere, the method enhances anomaly separability. Experiments on both unsupervised and semi-supervised settings demonstrate consistent improvements in FPR95 and AUROC over standard VAEs and related baselines.

**Strengths:**

1.The paper presents a well-motivated and theoretically sound reformulation of the VAE framework by introducing hyperspherical coordinates, effectively addressing the limitations of latent space concentration in high-dimensional settings.
2.Extensive experiments on both real-world and benchmark datasets show consistent and interpretable improvements in anomaly detection performance.

**Weaknesses:**

1.The writing and overall organization of the paper could be improved for clarity and coherence, figures and methodological explanations could be better integrated and more consistently formatted.
2.The paper would benefit from more comprehensive ablation studies to isolate the contribution of each design choice on anomaly detection performance.
3.As shown in Table 2, the proposed method does not reach state-of-the-art performance on several far-OOD datasets, suggesting that further validation or complementary analyses are needed to more convincingly demonstrate the effectiveness and generalizability of the approach.

**Questions:**

1.The paper would benefit from a more detailed complexity analysis, as the transformation from Cartesian to hyperspherical coordinates introduces additional computational steps and parameters.

---

> ### Author Response · Authors · 2025-11-20
> **"well-motivated and theoretically sound reformulation of the VAE"!**
>
> Weaknesses:
>
> 1. We would be happy to improve our manuscript if you have any specifics, noting the comment of reviewer #3: “Overall the paper is well-structured and easy to follow”. We will however work on editing the revised version for clarity.
>
>
> 2. The results presented in the paper already contain these ablations, though not presented in separate tables for lack of space. In both types of AD, we report the results for (V)AE+kNN, vMF-VAE+kNN, and our method. We also replaced kNN by another non-parametric method (IF).  Taking all of them together, they allow to assess each component: (V)AE+kNN vs vMF-VAE+kNN shows the effects of applying hypervolume compression vs not, while vMF-VAE+kNN vs our method, shows the effects of applying hypervolume compression using all hyperspherical coordinates vs using only one.
>
>
> 3.  Our method does reach state-of-the-art at the metric FPR95 (false positive rate at 95% true positive rate), which is key for anomaly detection for far-OOD in relation to comparable methods.
>
>
> Question:
>
> We provide in the supplementary a detailed measurement of the impact of the transformation in terms of the decreasing real processing speed, measured in batch/second (at a batch size = 200 samples), as the dimensionality increases.

---

### Author Response · Authors · 2025-12-03
**Message to the AC**

We are very disappointed by the quality of the reviews, which missed the whole point of the paper: does reduction in latent sparsity of the latent space of a VAE improve anomaly detection? $\rightarrow$ yes on actual use cases (not toy experiments, which are not informative). There are only a few minor good suggestions that we were very happy to consider for improving the manuscript.

Reviewer 1 comments were minor and all easily rebutted. They could hardly justify the rating (4). That reviewer even missed that our method reached state of the art, complaining that it does not!

Reviewer 2 is clearly not familiar neither with the literature (asking to justify the latent dimension, which is standard for the data and not the point of the paper), nor with the concept of high dimensional statistics (asking to describe volume concentrations and almost-orthogonality concepts within the paper instead of the current summary in appendix). Despite every critique being rebutted, reviewer 2 maintained a 4 rating without any more justification other than having read the other reviews...

And finally, reviewer 3 with a rating of 2, whose review is essentially a complaint that we should have written a different paper (on a different topic) and totally ignored our responses. Reviewer 3 is insisting that we use the methods and benchmarks from  ADBench, which is mostly not relevant for our paper (maybe ADBench is their own work since all the suggested extra references use that benchmark?).

We present a novel methodology that achieves state-of-the-art results on two public real-world datasets, as well as comparison on several standard benchmark datasets. Given these outcomes, the ratings (4,4,2) are difficult to comprehend.

---

### Meta-Review · Area_Chair_hcf8 · 2026-01-07

**Summary:**

The paper formulates the latent variables of a VAE using hyperspherical coordinates.  The main idea of the paper is to understand whether the reduction of sparsity in the latent space of a VAE improve anomaly detection.

The reviewers raised major concerns regarding the evaluation of the proposal against other baselines and datasets, and lack of explanations and comparisons against similar work.  Despite the authors' rebuttal, the concerns regarding the evaluation remain.  Thus, I recommend the rejection of the paper.

Strengths:
- Well motivated and theoretically sound VAE framework using hyperspherical coordinates
- Extensive experiments in un- and semi-supervised settings
- Strong appendix with additional intuition and background information

Weaknesses:
- Lack of comprehensive ablation studies to isolate the contribution of each design choice
- Lack of performance in OOD datasets
- Lack of statistical rigor in the evaluation
- Insufficient intuition or formal support on the methodological contributions
- Lack of convincing evidence that "equator clustering" is an issue in standard VAEs
- Incomplete related work and missing comparisons against similar work
- Missing large scale benchmarks
- Conceptual conflation: OOD detection treated as semi-supervised AD

**Reviewer Concerns:**

Reviewer 8nMQ raised concerns regarding the lack of performance in OOD datasets as well as lack of comprehensive studies to understand the contribution of each design choice.  The authors commented that they already reported ablations on the paper, and that the paper has good performance in FPR95 which is a key metric for anomaly detection.  While there were no specifics on which ablations were missing, I consider the reply from the authors sufficient.  However, the performance in OOD datasets is still limited.

Reviewer gFF6 raised concerns about the lack of statistical rigor in the evaluation, limited methodological accessibility, unclear evidence that "equator clustering" is an issue on standard VAEs.  While the authors provided answers in the rebuttal, there are still open questions given the limited answers (in particular related to the motivation).  The reviewer replied and quoted other issues raised by Reviewer whHn, and said that still leans toward reject.

Reviewer whHn raised concerns about the similarity with Ascarate et al. 2025, and the lack of discussion against related work on concentration of measure phenomena and compression, and, more importantly, the lack of large-scale benchmarks.  The authors provided answers to the reviewer in the rebuttal.  However, the reviewer mentioned that their concerns remained given that the proposal mainly is applying techniques from another area, and the lack of baselines as well as missing datasets.

**Reviewer Scores:**

Reviewer 8nMQ recommended a weak reject.  The authors partially addressed the reviewers concerns, by pointing out to the experiments already reported in the paper.

Reviewer gFF6 recommended a weak reject.  Despite the authors' rebuttal, the reviewer still maintained a negative assessment of the paper, and mentioned that they shared the issues raised by Reviewer whHn.

Reviewer whHn recommended a reject. And maintained their stance after the rebuttal since the authors didn't fully address the reviewer's concerns.

---

### Decision · Program_Chairs · 2026-01-26

Reject